# Decoupling of $\delta^{18}$O from surface temperature in Antarctica in an ensemble of Historical simulations

Sentia Goursaud Oger[1,2], Louise C. Sime[2], and Max Holloway[3]

[1]CEA, DAM, DIF, F-91297 Arpajon, France
[2]Ice Dynamics and Paleoclimate, British Antarctic Survey
[3]Scottish Association for Marine Science, Oban, UK

**Correspondence:** Sentia Goursaud Oger (sentia.oger@cea.fr)

**Abstract.** stable water isotopes recorded in Antarctic ice cores have traditionally been used to infer past surface air temperatures (SAT). During the historical period (1850 onward), observational data and good quality ice core records overlap, yielding an opportunity to investigate key relationships between ice core stable water isotope ($\delta^{18}$O) measurements and the Antarctic climate. We present a new ensemble of climate model simulations covering 1851-2004 using the UK Met Office HadCM3 general circulation model equipped with stable water isotopes. Our ensemble captures observed historical SAT and precipitation trends, and weak $\delta^{18}$O trends. The weak $\delta^{18}$O trends mean there is no significant relationship between SAT and $\delta^{18}$O over one third of Antarctica, and also half of our considered ice core sites, though relationships are stronger when using regional averages. The strongest regional relationships occur in the West Antarctic Ice Sheet (WAIS) region. This decoupling between SAT and $\delta^{18}$O occurs primarily because of the impact of autumnal sea ice loss during the simulated warming. The warming and sea ice loss is associated with: (i) changes in near-coastal air mass intrusions (synoptic effects) induced by changes in the large-scale circulation and/or sea ice; (ii) direct sea ice driven changes in moisture pathways (especially lengths) to Antarctica; and (iii) precipitation seasonality changes, again mostly driven by sea ice changes. Consequently when reconstructing temperatures over these timescales, changes in sea ice need to be considered; both to determine the most appropriate SAT and $\delta^{18}$O relationship, and to understand how uncertainties affect the inference of past temperature from ice cores $\delta^{18}$O measurements.

## 1   Introduction

Strong visible signs of Antarctic response to climate change have recently emerged. While a new sea ice cover minimum was recorded in February 2022 (Raphael and Handcock, 2022; Turner et al., 2022), with an extent of 1.97 million square kilometers, this record was broken the following year with sea ice extent falling to 1.91 million square kilometers on February 13th, 2023, associated with strong westerly winds and a 1.5 °C positive anomaly for Antarctic Peninsula air temperatures. The collapse of Antarctic ice shelves have similarly increased in frequency (Graham et al., 2022; Milillo et al., 2022; Wille et al., 2022). The consequent weakening of the buttressing force from the sea-ice free areas, and ice shelf collapse, acts to accelerate Antarctic ice loss. Thus warming of Antarctica will have significant consequences for global, and regional mean sea level (Edwards et al., 2021; Seroussi et al., 2020; Parsons et al., 2020; Garbe et al., 2020), alongside consequences for Antarctic life and its environs (Golledge et al., 2019; Post et al., 2019), which require thought about adaptation (Pörtner et al., 2022).

The relatively short satellite record (since 1979 only), and sparsity of in-situ observational data from Antarctica, mean that reconstruction of past temperature change is important for understanding natural variability, and hence our ability to detect anthropogenic climate change in Antarctica (Turner et al., 2004; Casado et al., 2023). Our understanding of pre-industrial climate change and its variability is mostly based on the reconstruction of temperature from proxy data (Pag, 2019). In Antarctica, stable water isotopes are the measurement most commonly used to reconstruct past surface air temperatures (SAT). This type of reconstruction is generally based on an empirical relationship between present day (PD) surface snow water isotopes and surface air temperature (Dansgaard, 1953). The relationship between SAT and the ratio of heavy water to light water isotopes (expressed as $\delta^{18}$O) from Antarctic surface snow is usually assumed to be linear. When this linear relationship is used to estimate past SAT values from ice core water isotope measurements, it is sometimes referred as the 'isotopic paleothermometer' (Lorius et al., 1969; Masson et al., 2000).

The isotopic paleothermometer approach has been successfully applied to deep ice cores to reconstruct past temperatures on long timescales (Jouzel et al., 2007; Lambert et al., 2008; Masson-Delmotte et al., 2010; Wolff et al., 2010). Casado et al. (2023) recently reconstructed the past 1000 year temperature record using an isotope to temperature conversion. This ice core based record was then used to show that the simulated temperature variability from the Atmospheric General Circulation (AGCM) models run in the frame of the Coupled Model Intercomparison Project Phases 5 (Taylor et al., 2012) and 6 (Eyring et al., 2016, CMIP6) is too low. The isotopic paleothermometer relationship has been shown to vary spatially over the Antarctic continent (e.g. Sime et al., 2008, 2009a). Goosse et al. (2012); Smerdon et al. (2015) and Neukom et al. (2018) show that noise and the spatial coverage of $\delta^{18}$O and other proxy data affect our understanding of these variations, while Goursaud et al. (2018, 2019) and others show smaller, or less reliable, changes in $\delta^{18}$O for a given temperature change in coastal regions (Isaksson and Karlén, 1994; Sime et al., 2008, 2009a; Abram et al., 2013; Thomas et al., 2013; Goursaud et al., 2017). Data to investigate SAT-$\delta^{18}$O relationships are sparse (Masson-Delmotte et al., 2008; Landais et al., 2017), nevertheless the PAGES 2k Network Antarctica2k (A2k) helped to address this question of geographical variations in the SAT-$\delta^{18}$O relationship, by defining regions and compiling the available $\delta^{18}$O data from ice cores (Stenni et al., 2017).

The geographical variability in the 'isotopic paleothermometer' is due to controls on $\delta^{18}$O other than related to SAT. These other controls include: changes related to atmospheric dynamics, such as changes in the synoptic and seasonal nature of precipitation (van Ommen and Morgan, 1997; Krinner and Werner, 2003; Jouzel et al., 2003; Sime et al., 2008; Servettaz et al., 2023) and air mass sources (Landais et al., 2021), various impacts from changes in Antarctic ice sheet morphology (Holloway et al., 2016; Werner et al., 2018; Buizert et al., 2021; Goursaud et al., 2021), and sea ice variability (Holloway et al., 2018; Cauquoin et al., 2023). The stability of the SAT-$\delta^{18}$O relationship has thus been of much of interest for more than two decades (Jouzel et al., 2003). Following Sime et al. (2008) and Sime et al. (2009a), the importance of changes in synoptic events, in the context of the anthropogenic warming, was recently explored by Wille et al. (2019) and Pohl et al. (2021), who confirm that the impact of synoptic changes on the SAT–$\delta^{18}$O relationship can be important for the past paleothermometer during warm climates (Dalaiden et al., 2020).

AGCMs equipped with stable water isotopes are a key tool to investigate the climate processes driving temporal variability in the paleothermometer relationship (e.g. Werner et al., 2001; Sime et al., 2008; Werner et al., 2018). For instance, AGCM

isotopic studies have focused on the effects of external forcing on the SAT–$\delta^{18}$O relationship, including elevation and greenhouse gases across a range of timescales (e.g. Sime et al., 2009b; Werner et al., 2018; Goursaud et al., 2021). A major result is that, for differing time-scales and driving mechanisms, different SAT–$\delta^{18}$O relationships can be obtained. This emphasises the importance of investigating the impact of atmospheric dynamical drivers, particularly changes in sea ice and precipitation seasonality during past warm periods in Antarctica (Sime et al., 2008, 2009b; Holloway et al., 2016). Only one AGM study has investigated the signature of stable water isotopes in Historical simulations (Yoshimura et al., 2008). While the applicability of the Antarctic paleothermometer relationship has been so investigated across various timescales, it has not yet been thoroughly investigated over the historical period. Furthermore, it has not been investigated using transient Historical (1851-2004) simulations.

Here, we run an ensemble of transient Historical (1851-2004) simulations, using the stable water isotope enabled coupled general circulation model, HadCM3. This ensemble provides a benchmark of Historical precipitated stable water isotopes covering the whole continent, and allows the SAT–$\delta^{18}$O relationship over the Historical period to be investigated. Firstly, we examine trends in SAT, precipitation and $\delta^{18}$O, and compare these against observed trends. We then examine SAT–$\delta^{18}$O relationships, including regional patterns, and the question of model dependence. Finally, in order to understand the drivers of $\delta^{18}$O change, we perform a detrended composite analysis for cold and warm year ensembles, and quantify the impact of seasonality changes in precipitation and $\delta^{18}$O. This analysis provides understanding of SAT–$\delta^{18}$O variability and, in particular, the role of sea ice change during the Historical period on this relationship.

## 2 Materials and Methods

### 2.1 Model and simulations

Here, we use the Hadley Center Atmosphere-Ocean general circulation model (HadCM3; AOGCM), to run six transient Historical simulations, HadCM3 is a version of the coupled Atmosphere-Ocean UK Met Office climate model (Pope et al., 2000; Gordon et al., 2000), which means that sea ice is prognostic. . The model is equipped with stable water isotopes (Tindall et al., 2009). Its horizontal resolution is 3.75° × 2.5°, and there are 19 vertical levels (Pope et al., 2000; Gordon et al., 2000; Tindall et al., 2009).

The setup of the Historical simulations is described in (Schurer et al., 2014), and follows the recommendations of the third Paleoclimate Modelling Intercomparison Project (PMIP3; Schmidt et al., 2011)(PMIP3; Schmidt et al. 2012). Each simulation is forced with time-varying orbital, solar, volcanic, land-use and well-mixed greenhouse gas forcing. As above, sea ice is not prescribed, rather calculated by the model. Changes in orbital parameters were calculated following (Berger, 1978). Volcanic forcing is that described in (Crowley et al., 2008). The solar forcing follows (Shapiro et al., 2011). Changes in $CO_2$, $N_2O$ and $CH_4$ were set following the PMIP3 standard (Schmidt et al., 2011). Changes in the abundances of 6 Halocarbons were prescribed following (Tett et al., 2007). Changes in land-cover were prescribed by reclassifying the Global land cover reconstruction developed by (Pongratz et al., 2008). Each of our simulations were only altered by starting each simulation a year apart.

We analyse HadCM3 surface air temperature (SAT, °C), precipitation (P, mm/month), precipitation weighted $\delta^{18}O$ ($\delta^{18}O$), and sea ice extent (defined as the region of ice-covered ocean, where the sea-ice concentration is >15%). HadCM3 provides a reasonable representation of Antarctic climate and $\delta^{18}O$ (Appendix A, as well as Turner et al., 2006; Tindall et al., 2009; Holloway et al., 2016).

## 2.2 Data and methods

We perform a model-data comparison using the Stenni et al. (2017) ice core data compiled by the PAGES A2k project by binning our model output, including $\delta^{18}O$, into 5-years equivalent averages and compute anomalies relative to the 1960-1990 mean. In order to investigate the historical mean climate state and variability, we compute ensemble mean values, covering the period 1851-2004 using monthly outputs. Trends over the whole period, as well as for the last 50 years, are calculated using linear regressions. Where we regress climate variables against $\delta^{18}O$, the linear regressions are computed using the stacked individual ensemble members, rather than using the ensemble mean. This approach ensures that the ensemble variability is included in our linear regression statistics and increased the number of points on which the regressions are processed. Gradients from the linear regressions are provided with a plus/minus standard error. Results from linear relationships are stated only where they are significant, using a p-value $\leq 0.05$.

Our Historical SAT–$\delta^{18}O$ linear relationship at the regional scale are compared with the regional slopes and correlation coefficients that we computed from the AGCM ECHAM6-wiso equipped with water stable isotopes (Cauquoin et al., 2019). The water stable module of this last generation of the model ECHAM was updated compared to its predecessor, especially (i) the supersaturation parameters, (ii) the kinetic fractionation at the evaporation over oceans, now assumed to be independent of the wind speed in order to better represent the d-excess versus deuterium relationship from the Antarctic Snow reported by (Masson-Delmotte et al., 2008), and finally (iii) the sublimation processes now accounting for the isotopic content of snow over sea ice. Here, we use a simulation run at a T127L95 resolution ( 0.9° x 0.9° horizontal resolution and 95 vertical levels) and nudged towards the ERA5 reanalyses (Hersbach et al., 2020) over the period 1979 – 2022 Cauquoin and Werner (2021).

Composites are used to interpret our results. Warm and cold (versus mean) mean annual composites results are defined using detrended annual area-weighted SAT. The years of the ensemble mean with SAT below (above) the mean minus (plus) one standard deviation constitute a cold (warm) ensemble.

To examine the impact of changing seasonality over the Historical period, we isolate the impact of precipitation and $\delta^{18}O$ seasonal changes, recorded in the ensemble mean, on the precipitation weighted $\delta^{18}O$ between the first 50 years of the simulation and the last 50 years of the simulations (c.f. Liu and Battisti, 2015; Holloway et al., 2016; Sime et al., 2019). This is calculated as:

$$\Delta^{18}O_{seas} = \frac{\sum\limits_{j} \delta^{18}O_j^{recent} \times P_j}{\sum\limits_{j} P_j} - \frac{\sum\limits_{j} \delta^{18}O_j \times P_j}{\sum\limits_{j} P_j} \tag{1}$$

$$P_{seas} = \frac{\sum\limits_{j} \delta^{18}O_j \times P_j^{recent}}{\sum\limits_{j} P_j^{recent}} - \frac{\sum\limits_{j} \delta^{18}O_j \times P_j}{\sum\limits_{j} P_j} \tag{2}$$

The summations, with index j, are over the 12 months of the year. Variables with superscript "recent" indicate that they
were extracted for the last 50 years of the simulation whereas the variables without superscript indicate that the variables were
extracted for the first 50 years of the simulation.

## 3 Results

This section uses these model data and methods to examine: trends in Antarctic SAT, precipitation, sea ice and $\delta^{18}$O, including
at the continental and regional scale; relationships between temperature versus $\delta^{18}$O, including their stability, and model
dependency; and finally, the drivers of $\delta^{18}$O changes.

### 3.1 Trends in Antarctic SAT, precipitation, sea ice and $\delta^{18}$O

We analyse our simulations against available observations and reanalysis data (e.g. the Climate Forecast System Reanalysis
and the the National Centers for Environmental Prediction reanalyses 2) at the Antarctic wide-scale before evaluating regional
scale climate changes. This is followed by a comparison of simulated $\delta^{18}$O changes against the Stenni et al. (2017) ice core
dataset. We note that all trends outlined below are similar regardless of whether we use the full Historical period, or the last
50-years of the ensemble simulations. Nevertheless, to permit the most direct comparison where possible, we have matched
our calculations to the periods used by other authors.

#### 3.1.1 Continental trends in climate

Our simulated SAT trend over Antarctica is 0.12±0.02 °C per decade over the last 50 years (Figure 1). This is consistent with
observations of 0.12±0.07 °C per decade over 1957-2006 (Steig et al., 2009) and 0.11±0.08 °C per decade over 1959-2012
(Nicolas and Bromwich, 2014). Our trend is lower than the 0.22±0.04 °C trend of Casado et al. (2023), however this is itself
dependent on their isotope-to-temperature reconstruction method. Over East Antarctica, our simulated Historical SAT trend is
weaker (0.10±0.02 °C per decade; r=0.59) than over the West Antarctica (0.15±0.02 °C per decade; r=0.75). This compares
with 0.10±0.07 °C per decade for the East and 0.17±0.06 °C per decade for the West calculated by Steig et al. (2009).

Our simulated Historical precipitation trend over the last 50 years is 3.1 mm/y per decade (Figure 1), which lies between the,
admittedly wide, Bromwich et al. (2011) reanalyses based equivalent values of 0.4±1.8 mm/y per decade from the Climate
Forecast System Reanalysis (CFSR), and 7.1±1.5 mm/y per decade from the National Centers for Environmental Prediction
reanalyses 2 (NCEP-2) over 1979-2009. The HadCM3 results are also is agreement with Dalaiden et al. (2020) who show an
increase in both WAIS and EAIS simulated precipitation.

Our HadCM3 simulated Historical Antarctic September sea ice decrease is $-0.20\pm0.01$ x $10^6$ km$^2$/decade over the period 1851-2004 (r=-0.69), and $-0.40\pm0.06$ x $10^6$ km$^2$/decade over the period 1954–2004 (r=0.67). This is consistent with the recent results of Shu et al. (2020), who calculated Antarctic September sea ice trends of -0.45 and -0.43 x $10^6$ km$^2$/decade during the period 1979–2005 from the Coupled Model Intercomparison Project (CMIP) 5 and 6 model results, respectively. Whilst neither the Shu et al. (2020) nor our HadCM3 sea ice trends match the observed slope of +0.10 x $10^6$ km$^2$/decade (p=0.16)

(Shu et al., 2020), they do agree with each other. It is also worth noting that, since 2016, Antarctic sea ice has begun losing significant area during both summer and winter.

     The simulated Historical changes in Antarctic-wide SAT and precipitation and sea ice thus seem in reasonable agreement with other model results, and possibly also observations (though this is less clear). Trends, however, vary between regions.

### 3.1.2   Regional-scale trends

We now look at the simulated climate, using the Antarctic regions defined by the PAGES A2k community (see Figure 2, regions defined by Stenni et al., 2017). Every Antarctic region shows a simulated increase in SAT over the Historical period (1851–2004, Figure 2, Table A1), but these vary across the continent. Warming trends are strongest for the Peninsula and Dronning Maud Land regions (0.11 °C per decade, and r≥0.8), closely followed by the WAIS and the Weddell coast regions (0.08 °C per decade, and r≥0.8). The Plateau and the Indian coast show weaker warming trends of 0.04 and 0.05 °C per decade, respectively

(r=0.6 for both). These historical trends approximately match Turner et al. (2020) results derived from station data.

     Our simulated SAT trends resemble the Stenni et al. (2017) regional warming trends (Table A1). Stenni et al. (2017) found cooling trends for the Plateau, the Weddell coast, and Victoria Land (from -0.13 to -0.05 °C per decade), where we found a small warming; they found a larger trend for the Peninsula (from 0.2 to 0.29 °C per decade), though again we point out that the Stenni et al. (2017) values, like the Casado et al. (2023) results, are partly dependent on isotope-to-temperature reconstruction

methods. At the scale of station locations, Jones et al. (2019) also show the highest trends for the Peninsula.

     Similar to SAT, HadCM3 shows an increase in precipitation for all the Antarctic regions, both for the full Historical period, and for the last 50 years. Similar to temperature, the Peninsula features the strongest trend of 7.8 mm/y per decade over the Historical period, whereas the Plateau and the Ross sections display weaker trends of 0.5 and 0.82 mm/y per decade over the same period. Interestingly, Thomas et al. (2017) and Medley and Thomas (2019) found similar results. Whilst these results are

175 from ice cores, these are not dependent on the interpretation of isotopes, instead they mostly use profiles of density and layer counting from relevant age markers (e.g. chemical species, radio isotopes, biologically compounds).

     For sea ice, HadCM3 simulates a sea ice decrease around all sectors of Antarctica, except for Weddell sector (Table B1). Trends are largest in the Indian sector ($-49\pm5$ x $10^3$ km$^2$ per decade; r=-0.67) and smallest for the Pacific sector ($-25\pm4$ x $10^3$ km$^2$ per decade r=-0.44). Whilst these results are not compatible with pre-2016 satellite observations, they are consistent with

180 other climate simulations (Shu et al., 2020).

     Despite the simulated increases in SAT and precipitation, $\delta^{18}$O simulated by HadCM3 shows a very weak trend of $0.04\pm0.003$ ‰ per decade (r=0.21) over the last 50 years. Interestingly, Casado et al. (2023) provide a higher trend from 1950–2005 of $0.11\pm0.02$ ‰ per decade, based on ice core data. Different reasons could explain that mismatch that we are not able to elucidate

so far, inter alia: (i) a model discrepancy to resolve processes, (ii) the model resolution, (iii) the geographical distribution of the ice core locations, (iv) the different methods for the SAT – $\delta^{18}$O calibration. Section 5 of this paper focuses on investigating and explaining the $\delta^{18}$O trends. Before this, we provide a brief overview of the regional picture.

At the regional scale, over the Historical period, trends are small (Figure 2). In terms of linear relationship, it is null for the Victoria Land, while the gradient is the highest for the Weddell coast with a trend of 0.05 ‰ per decade (r=0.39), and the correlation coefficient is the highest (e.g. the strongest linear relationship) for the peninsula with a trend of 0.04 ‰ per decade (r=0.57). Over the last fifty years, only three regions, the Indian, the Weddell and Dronning Maud Land coastal regions keep on displaying significant $\delta^{18}$O trends, that double or more compared to the Historical period, with gradients of 0.08, 0.08 and 0.14 ‰ per decade respectively. Stenni et al. (2017) made a $\delta^{18}$O trend statistics based on ice core anomalies using unweighted composites over the period 1900-2000, based on 5-years bins. They found only 3 regions with significant trends, which are the Indian coast, the peninsula and Dronning Maud land, with gradients in the range of our results, while higher for the peninsula and Dronning Maud land (mean trends of 0.15 ‰ per decade and 0.11 ‰ per decade respectively). Comparatively, Casado et al. (2023) calculated trends over windows varying between 35 and 60 years and using a persistence method. They found gradients with the same range of values, from 0.09 ‰ for the Indian coast, to 0.19 ‰ for the Weddell coast, while they found significant relationships where we do not, for time windows varying from 40 to 65 years. Note that for most of the regions, the significance of our simulated relationships disappear for time windows shorter than 75 years (Appendix D). This could be explained either by the simulated anthropogenic variability being too low, as suggested by (Casado et al., 2023), or a change of the drivers on $\delta^{18}$O. The disparities between our results and the previous studies could be explained by the different time windows, the different methodologies, the lack of ice core data to make representative regional reconstructions, or a model discrepancy. While Casado et al. (2023) carefully investigated the impact of the data stack method and the time-window on the $\delta^{18}$O reported trends, we suggest that an extended study could compare the statistical and dynamical methods on both ice core data and water stable isotope enabled AGCM outputs to complete the analysis.

## 3.2 Temperature versus $\delta^{18}$O relationships

Given that much of ice core science is underpinned by the relationship between temperature and $\delta^{18}$O (Jouzel et al., 2003), and having discussed simulated SAT, precipitation, sea ice and $\delta^{18}$O trends, we now investigate temperature versus $\delta^{18}$O relationships. Sime et al. (2008) demonstrate that there is no clear relationship between the spatial versus temporal SAT-$\delta^{18}$O gradients across Antarctica. We therefore focus on comparing the last 50 years of our simulations against the whole Historical period. To enable a consideration of model dependency, we also compare our Historical ensemble against a nudged ECHAM6-wiso simulation (Table 1).

### 3.2.1 Antarctic-wide and regional scale

The simulated SAT-$\delta^{18}$O relationship, calculated using annual means on each grid point, is statistically significant over 66% of Antarctica (Figure 3). For the continent as a whole, we simulate a mean Antarctic SAT-$\delta^{18}$O gradient of 0.57±0.06 ‰/°C (r=0.62) over the Historical period, increasing to 0.67±0.13 (r=0.60) over the last 50 years of the simulation. The comparable

numbers for an Antarctic-wide SAT-$\delta^{18}$O relationship in Casado et al. (2023) are 0.49 to 0.69 ‰/°C. In our simulations, non significant SAT-$\delta^{18}$O relationships occur in: the eastern Antarctica between 40 and 100° E; all regions between 140 and 220° E covering the Wilkes coast, Victoria Land and some parts of Queen Mary Land; and the coast of Dronning Maud Land with some areas at 350-360° E joining the South Pole. Non significant relationships were also reported in observations and model outputs. For instance, Goursaud et al. (2018) report no SAT-$\delta^{18}$O relationship at the annual scale over the coast of Dronning Maud Land, the Victoria Land, some of the Indian coast and the Peninsula. An absence of SAT-$\delta^{18}$O relationship derived from firn/ice cores were also published (e.g. Goursaud et al., 2019; Bertler et al., 2011; Vega et al., 2016; Goursaud et al., 2017). More recently, Casado et al. (2023) also found no significant relationships, though possibly because of a lack of data, for some regions. For them, these regions were the Indian and Weddell coasts and Victoria Land. (Please see section 3.3 for a discussion of why these regions do not show statistically significant temperature versus $\delta^{18}$O relationships). In contrast, the Peninsula, part of the WAIS coast, as well as some parts of the Plateau show much stronger SAT-$\delta^{18}$O relationships. Indeed, some coastal areas are associated with the highest correlation coefficients, ranging between 0.15 and 0.45. The highest SAT-$\delta^{18}$O gradients in the Plateau region, as well as south of the Filchner ice shelf, can exceed 0.75 ‰/°C. Casado et al. (2023) also found the largest SAT-$\delta^{18}$O gradients in similar regions.

This Antarctic-wide picture of geographical variability in the temperature versus $\delta^{18}$O relationship is reasonably consistent with previous studies covering parts of the Historical period (Sime et al., 2009a; Stenni et al., 2017; Goursaud et al., 2019), as well as measurements from coastal firn cores (Isaksson and Karlén, 1994; Abram et al., 2013; Thomas et al., 2013; Goursaud et al., 2017). Interestingly, Guan et al. (2016, 2020) associate similar results (low or negative SAT-$\delta^{18}$O relationships) with higher source temperatures. Here, we suggest this is mainly driven by sea ice retreat (See section 3.3).

### 3.2.2   Stability over the Historical period and model dependency

Results from HadCM3 are similar for both the last 50 years and the whole Historical period (Table 1): SAT-$\delta^{18}$O gradients vary between 0.3 to 0.7 ‰/°C. The average difference is 0.07 ‰/°C. The only region with a statistically different result is Dronning Maud Land, with SAT–$\delta^{18}$O gradients of 0.76$\pm$0.12 ‰/°C and 0.49$\pm$0.05 ‰/°C over the last 50 years and the whole Historical period, respectively. Thus, for most of the continent, our HadCM3 results over the last 50 years and the whole Historical period appear equivalent.

Interestingly, the ECHAM6-wiso SAT-$\delta^{18}$O gradients calculated here are on average twice lower than those computed using the ECHAM5-wiso simulations nudged to ERA-interim over the period 1979–2013 and published in Stenni et al. (2017). Thus, the improvements to the ECHAM6 coding seem to bring HadCM3 and ECHAM into alignment (Table 1). We consider these new ECHAM6-wiso and HadCM3 values to be likely more accurate. We can note the small differences between HadCM3 and ECHAM6-wiso: ECHAM6-wiso simulates slightly stronger relationships with a mean correlation coefficient difference of 0.04, while gradients tend to be slightly higher in HadCM3 with a gradient difference of 0.13 ‰/°C. The only notable differences are for Dronning Maud Land and the Indian coast with stronger relationships and higher gradients simulated by HadCM3. Thus, whilst it is unclear whether the nudging of ECHAM6-wiso towards ERA5 reanalysis, the model resolutions, the model physics or differences in sea ice behaviours, are the main reason for these discrepancies, it is clear that simulated

temperature versus $\delta^{18}$O relationships have low but significant uncertainties. These need to be considered, both regionally and for the most relevant climate state, before being undertaking any inferences of past temperatures using isotopes measured in ice cores.

## 3.3 Drivers of $\delta^{18}$O changes

We use two approaches to investigate the mechanisms driving simulated $\delta^{18}$O changes. First, we separate and compare extreme warm and cold years both for annual (Figure 4, Table C1) and seasonal (Figure 5) data by generating (annual and seasonal) composites with mean annual Antarctic SAT anomalies greater than plus or minus two standard deviations from the mean, respectively. Second, we isolate the impact of changing precipitation seasonality on $\delta^{18}$O, showing simple months values (Figure 6) and also following the decomposition method used in Liu and Battisti (2015); Holloway et al. (2016) and Sime et al.
(2019) (Figure 7).

As expected, the spatial patterns of SAT, and sea ice anomalies tend to vary together, with the pattern is approximately mirrored between cold and warm composites (Figure 4, top and bottom panels, respectively). Whilst fully isolating the drivers of $\delta^{18}$O is tricky, together Fig 4 to 7 suggest that the primary mechanism driving continental-scale SAT-$\delta^{18}$O decoupling in HadCM3 is the simulated loss of sea ice over the historical period (Figure 5dh).

The September average sea ice area across the warm composite is 5.8 x10$^6$ km$^2$ less than in the cold composite. Given that this reduction occurs primarily during winter (Figure 5c; there is almost no summertime sea ice around Antarctica), warmer years tend to receive relatively more precipitation during winter months compared to cold years, partially offsetting the warming signal in $\delta^{18}$O. This can be seen in Figure 5, displaying seasonal anomalies (for the winter season, e.g. from June to August, and for the summer season, e.g. from December to February) in precipitation, $\delta^{18}$O and sea ice between the warm and
cold composites: the largest (smallest) precipitation and $\delta^{18}$O anomalies occur during winter (summer) months. Precipitation anomalies peak in autumn and winter, whilst $\delta^{18}$O anomalies peak in winter and spring (Figure 6), the latter coincident with the annual maximum sea ice extent and largest sea ice area anomalies. The relative increase in winter precipitation during warm years acts to reduce $\delta^{18}$O across Antarctica, compared to if the seasonality of precipitation remained unchanged. This is perhaps clearest seen in Fig. 7, where Fig. 7a is predominantly blue - which says that precipitation seasonality changes
are acting to decrease $\delta^{18}$O. The effect of changing seasonality is particularly large in the Indian, Dronning Maud Land and Victoria Land (through the Wilkes Land) sectors, which are prone to air mass intrusions (Fig. 5c and 7a).

Although the reduction in sea ice area simulated in the warm composite and throughout the Historical ensemble increases the proportion of winter precipitation, negatively influencing $\delta^{18}$O, less sea ice also shortens the distance between evaporation source and precipitation site, which has an opposing positive influence on $\delta^{18}$O. This effect is evident as negative sea ice
anomalies adjacent to coastal regions with large positive $\delta^{18}$O anomalies in Figure 7b as well as in Figures 5cf. Sea ice loss may also allow locally sourced precipitation to penetrate further inland into the Antarctic interior, which usually receives precipitation sourced from lower latitudes (Gao et al., 2023), as well as promoting an increase in high intensity precipitation events during cold seasons (e.g. Schlosser et al., 2004). Consequently, sea ice loss during the Historical period leads to competing influences on $\delta^{18}$O and considerable spatial variability in SAT-$\delta^{18}$O relationships (Figure 3a), with no significant relationship

in several regions. Changes in the precipitation seasonality (Figure 7a) reduce mean Antarctic $\delta^{18}O$ by 0.16±0.19 ‰ and for the lowest changes by 0.11 ‰ over the Plateau and no changes over the Peninsula. Changes in the seasonal cycle of $\delta^{18}$O (Figure 7b) are more spatially variable and largely sea-ice driven, with a depletion up to -0.45 ‰ over the Plateau and Victoria Land and an enrichment from 0.1 to 1.0 ‰ in coastal regions.

These Historical simulations indicate that Antarctic $\delta^{18}$O is highly sensitive to patterns of sea ice change, which influence atmospheric dynamics, air mass pathways and lengths, and water isotope evaporation and condensation temperatures. Our results are particularly sensitive to autumnal sea ice changes: the largest simulated reduction in sea ice occurs in Autumn, coincident with the largest changes in $\delta^{18}$O (Figure 5c). The dynamic processes behind the sea ice extent induced $\delta^{18}$O changes are complex and multiple. Although the Southern Annular Mode, leading mode of the atmospheric variability in the Southern Hemisphere, might explain part of these $\delta^{18}$O simulated changes (Appendix G), a more comprehensive study might investigate the impact of the atmospheric circulation changes.

## 4 Conclusions

Results from six transient simulations over the Historical period allow us to examine Antarctic precipitation weighted $\delta^{18}$O and its relationship with Antarctic SAT. The ensemble features a rise of mean-Antarctic SAT of 0.12±0.02 °C/decade over the last 50 years, consistent with previous studies Steig et al. (2009); Nicolas and Bromwich (2014). In agreement with observations, the associated simulated trend in the water isotope $\delta^{18}$O is weak (+0.04±0.03 ‰/decade). Unlike SAT, the $\delta^{18}$O trend is weaker during the last 50 years, compared to the complete Historical period. This implies that over the last fifty years of our ensemble, $\delta^{18}$O is influenced by processes other than SAT-related condensation temperature, *i.e.* ice and atmospheric processes oppose purely SAT and condensation temperature controls on $\delta^{18}$O.

Although the consequences of these competing effects vary spatially, they result in non-statistically significant relationships between SAT and $\delta^{18}$O for approximately one third of the continent over the Historical ensemble. Non-significant SAT-$\delta^{18}$O relationships occur across much of Antarctica between 40 and 100° E; the Wilkes coast, Victoria Land, part of Queen Mary Land; the coast of Dronning Maud Land, and areas of the South Pole region. Interestingly, we find similar but slightly weaker SAT-$\delta^{18}$O correlations and slightly higher gradients compared to ERA5–nudged ECHAM6-wiso simulations at the regional scale.

We suggest three processes that lead to weak SAT-$\delta^{18}$O relationships during the Historical ensemble. Firstly, Historical changes in near-coastal air mass intrusions, induced by changes in the large-scale circulation and/or changes in synoptic events, have an impact on both the moisture source and precipitation regime. Secondly, changes in sea ice concentration impact moisture pathways to Antarctica. And thirdly, changes in precipitation seasonality, largely driven by sea ice changes, biases the relative impact of cold vs warm season precipitation on precipitation-weighted $\delta^{18}$O. Fyke et al. (2017) show similar spatial patterns in Antarctic precipitation during the pre-industrial (Fyke et al. (2017), Fig 2.c), driven by the large-scale circulation and acting to regulate atmospheric moisture transport and regional sea ice variability (Fyke et al., 2017; Marshall et al., 2017; Raphael et al., 2019). Changes in Antarctic sea ice (Holloway et al., 2016) and moisture source changes (Landais et al., 2021)

have also been proposed as responsible for driving the Last Interglacial $\delta^{18}$O peak. Additionally, Sime et al. (2008) show that (sea ice related) changes in the seasonal and synoptic distribution of Antarctic precipitation were jointly responsible for a partial decoupling between SAT and $\delta^{18}$O. These results support the role of environmental changes, particularly sea ice, generating significant variability in the Antarctic SAT–$\delta^{18}$O relationship.

In conclusion, the results from this isotope-enabled Historical ensemble permit investigation of the SAT–$\delta^{18}$O relationship during the Historical period and the mechanisms driving its spatial variability. Whilst we recognise the limitations, in terms of spatial resolution and a simplistic sea ice sub-model, of our chosen model, HadCM3, we value the ability to perform a several-member ensemble of >100-year simulations using a fully coupled, isotope-enabled model.

For future work, higher resolution and more physically advanced models could be used, alongside new tracer methods (Gao et al., 2023). Firn model could also be used to examine the interesting question of post-deposition effects such as the redistribution of snow by the wind (Libois et al., 2014), snow-vapour exchanges (Casado et al., 2016; Ritter et al., 2016), and snow metamorphism (Casado et al., 2021) on Historical $\delta^{18}$O. Finally, more stable water isotope records from Antarctic ice and firn core data are more than needed to evaluate models, as well as to lead model-data investigations of past climates, comparing SAT–$\delta^{18}$O relationships from different water stable isotopes enabled model, in line with the work of the Stable Water Isotope Intercomparaison Group 2 (SWING) (Risi et al., 2012).

*Code and data availability.*   The code and the simulation outputs can be made available on request.

*Author contributions.*   SGO and LCS co-designed the study. MH ran all simulations. SGO conducted all analysis and produced all figures. SGO and LCS co-wrote the first draft of the manuscript. All authors contributed to the final manuscript version.

*Competing interests.*   The authors declare that they have no conflict of interest.

*Acknowledgements.*   We thank Alexandre Cauquoin for proving the ECHAM6-wiso outputs used in this manuscript. This work has received support from the NERC National Capability International grant SURface FluxEs In AnTarctica (SURFEIT): NE/X009319/1. LCS acknowledges additional support from DEEPICE: Understanding Deep Ice Core Proxies to Infer Past Antarctic Climate Dynamics, EU-H2020 G.N.955750; ANTSIE: ANTarctic Sea Ice Evolution from a novel biological archive: EU-H2020 G.N.864637 and TiPES: Tipping Points in the Earths System: EU-H2020 G.N.820970. Simulations were run and analysed on NERC's ARCHER2 and JASMIN platforms.

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

# Table

**Table 1. Historical SAT–$\delta^{18}$O relationships at the regional scale.** Slope (in ‰/°C) plus or minus the standard error, and the correlation coefficient (into brackets) of the surface-weighed average of surface air temperature against the surface-weighed average of $\delta^{18}$O for the Antarctic regions as defined in the PAGES Antarctica2k project (Stenni et al., 2017): the plateau, the Indian coast, the Weddell coast, the Peninsula, the WAIS, Victoria Land and Dronning Maud Land, simulated by the ECHAM6-wiso model (, over the period 1979-2022, 44 points, 'ECHAM6-wiso') and simulated by HadCM3 over he last 50 years (1955-2004, 50 points, 'last 50 years of HadCM3'), and over the whole historical simulated period (1851-2004, 154 points, 'Historical HadCM3') using the ensemble mean of the six simulations (see methods). All the relationships are significant (p-values<0.05).

| | ECHAM6-wiso | last 50 years of HadCM3 | Historical HadCM3 |
|---|---|---|---|
| Plateau | 0.48±0.07 [0.71] | 0.61±0.14 [0.52] | 0.57±0.07 [0.53] |
| Indian coast | 0.29±0.08 [0.48] | 0.55±0.15 [0.46] | 0.67±0.07 [0.59] |
| Weddell coast | 0.49±0.11 [0.57] | 0.57±0.11 [0.59] | 0.57±0.07 [0.57] |
| Peninsula | 0.37±0.05 [0.74] | 0.28±0.06 [0.52] | 0.31±0.02 [0.71] |
| WAIS | 0.56±0.07 [0.75] | 0.60±0.12 [0.58] | 0.50±0.05 [0.61] |
| Victoria Land | 0.43±0.13 [0.46] | - | 0.30±0.12 [0.19] |
| Dronning Maud Land | 0.43±0.13 [0.46] | 0.76±0.12 [0.69] | 0.49±0.05 [0.60] |
| West Antarctica | 0.49±0.11 [0.59] | 0.50±0.10 [0.57] | 0.70±0.07 [0.62] |
| East Antarctica | 0.48±0.08 [0.69] | 0.49±0.10 [0.57] | 0.56±0.06 [0.58] |
| All Antarctica | 0.45±0.09 [0.59] | 0.67±0.13 [0.60] | 0.57±0.06 [0.62] |

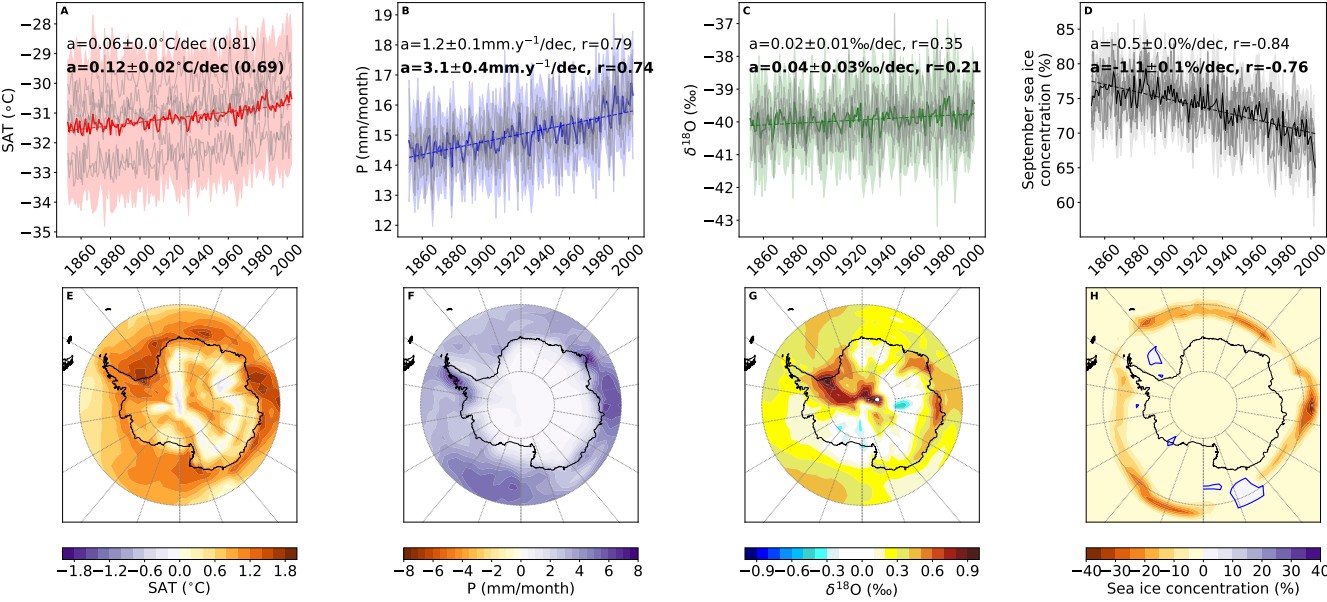

**Figure 1. Antarctic trends over the periods 1851-2004 and 1955-2004.** Each column is associated to a climate variable: the first column for the (**A, E**) surface air temperature ("SAT" in °C, in red), the second column for the (**B, F**) precipitation ("P in mm/month", in blue), (**C, G**) $\delta^{18}O$ (in ‰, in green) and (**D, H**) sea-ice concentration ("SIC" in %, in grey). The first row displays the time series of Antarctic surface weighted averages over the period 1851–2004. Colored solid lines represent the annual average of the ensemble mean, the colored surfaces represent the annual standard deviations of the ensemble mean, the grey solid lines the simulations, and the dashed lines, the linear regressions. The slopes ("a") plus or minus the standard errors, as well as the correlation coefficients ("r") are given on the top right of the figures, with the first row corresponding to the whole historical period 1851–2004, and the second row, in bold, over the last 50 years 1955–2004. The second row displays the map of anomalies over the periods 1955–2004 against 1854–1904.

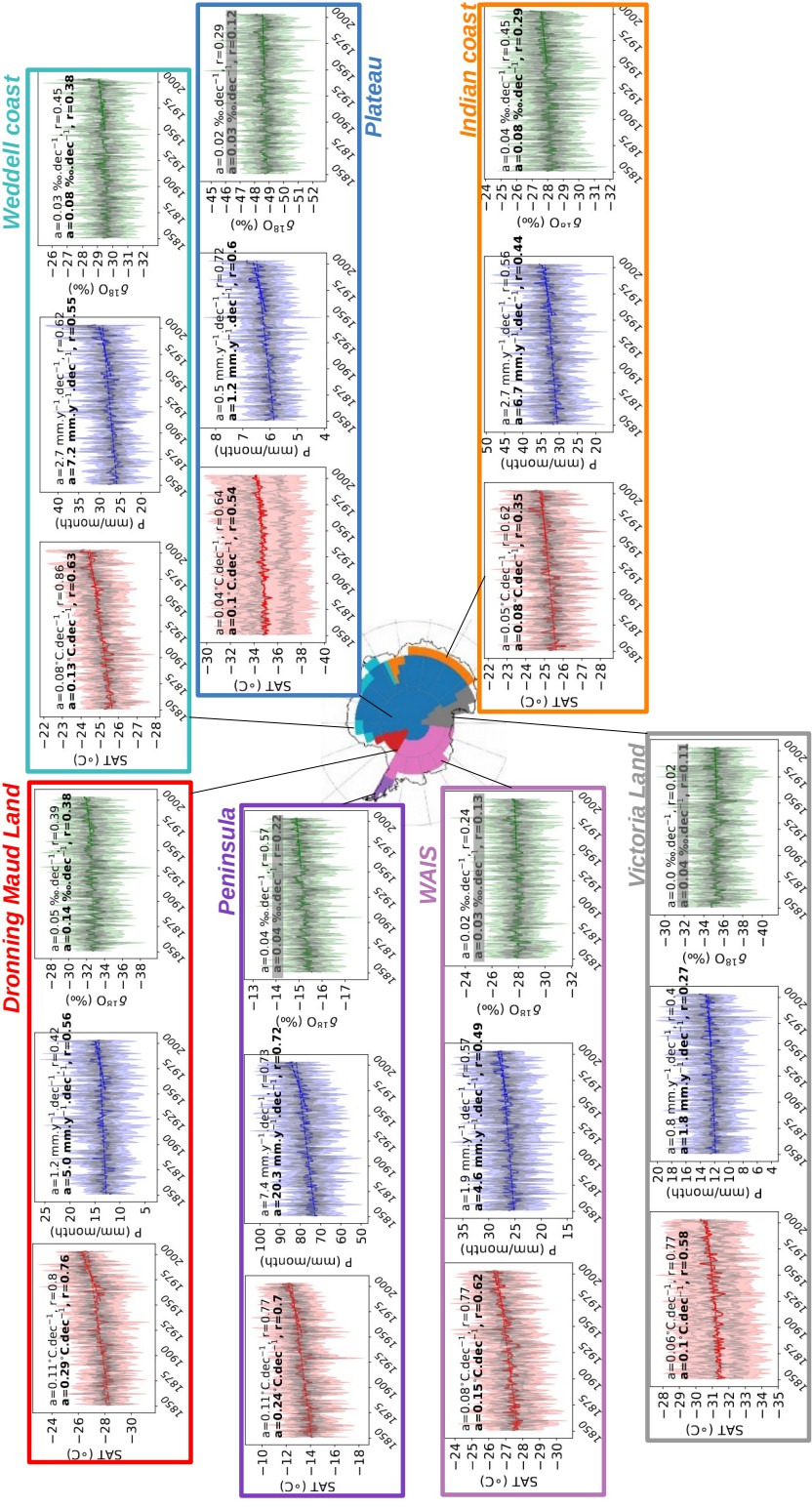

**Figure 2. Regional trends over the periods 1851-2005 and 1955-2005.** Time series of surface weighted averages of (**A**) surface air temperature ("SAT" in °C, in red), (**B**) precipitation ("P in mm/month", in blue), (**C**) $\delta^{18}O$ (in ‰, in green) and (**D**) sea-ice concentration ("SIC" in %, in grey) over the period 1851–2005, for the different Antarctic regions as defined by (Stenni et al., 2017). Colored solid lines represent the annual average of the ensemble mean, the colored surfaces represent the annual standard deviations of the ensemble mean, the grey solid lines the simulations, and the dashed lines, the linear regressions. The slopes ("a") and the correlation coefficients ("r") are given on the top right of the figures. The first row reports the regional slope and correlation coefficient over the whole historical period, while the second row reports he regional slope and correlation coefficient over the last simulated years. Grey shaded rows correspond to non significant relationships (p-value>0.05).

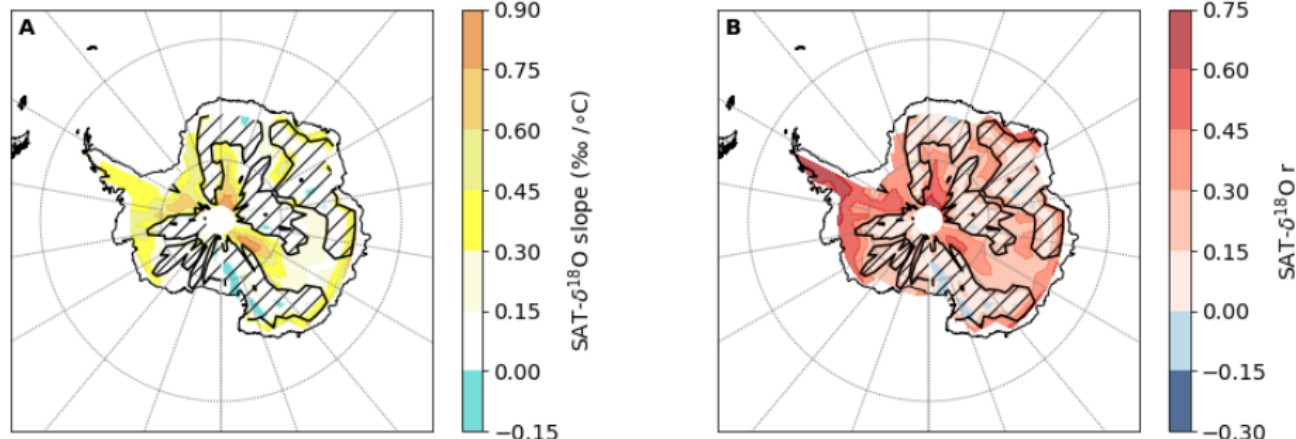

**Figure 3. Historical SAT–$\delta^{18}$O pattern.** The SAT-$\delta^{18}$O slopes (left panel) and correlation coefficients (right panel) simulated by the HadCM3 over the historical period at the interannual scale at each grid point. Gradients (in ‰/°C, "A") and correlation coefficients ("r") of the SAT–$\delta^{18}$O relationships at the interannual scale (151 points) at each Antarctic grid point and based on the stacked simulations. Regions of hashed black lines indicate to no significant relationships (p-value $\geq$0.05).

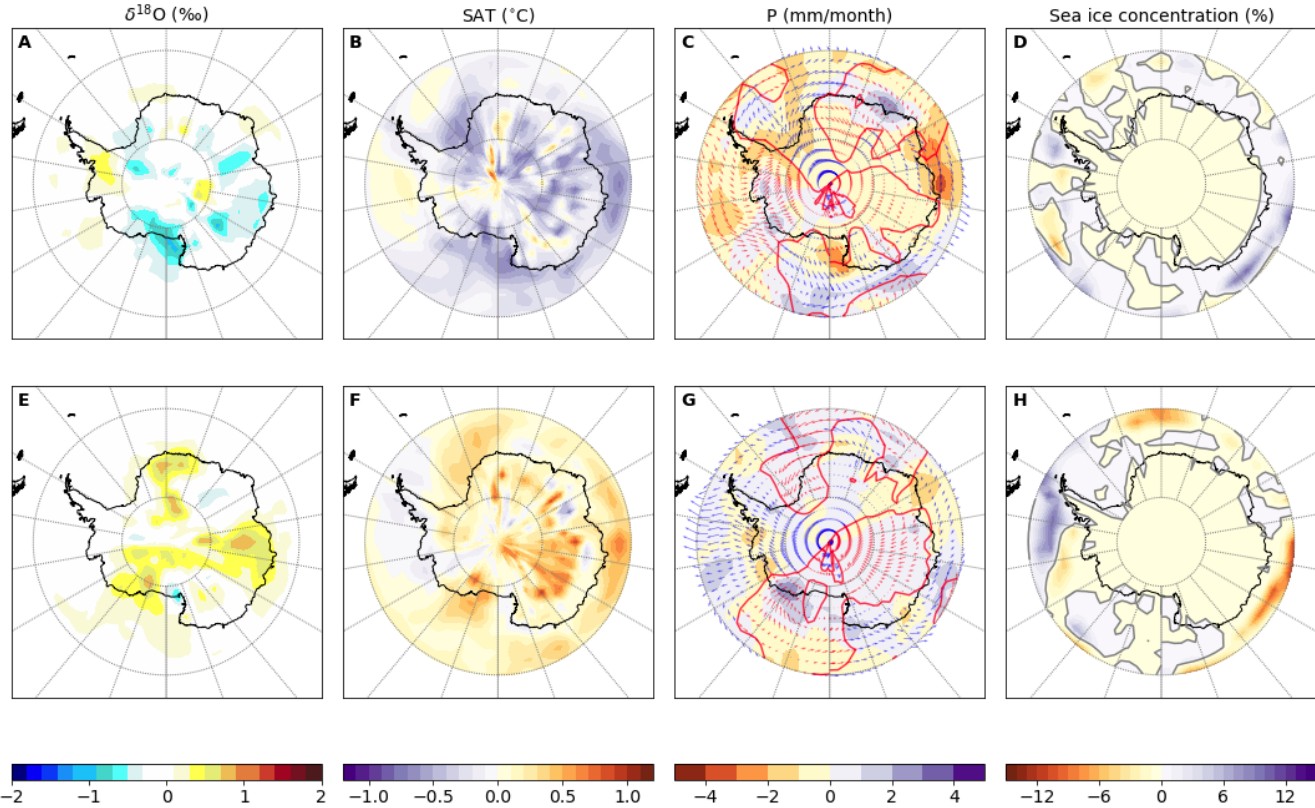

**Figure 4. Composite maps for cold (top) and warm (bottom) years.** Maps of Antarctic (**A, E**) $\delta^{18}O$ (in ‰), (**B, F**) surface air temperature ("SAT" in °C), (**C, G**) precipitation ("P" in mm/month), and (**D, H**) sea-ice concentration ("SIC" in %), for years with surface air temperatures below (top row) and above (bottom row) two standard deviations in the ensemble mean over the period 1851-2004. Maps use the detrended ensemble mean (See Methods). There are 8 years (out of the 155 simulated years) with SAT anomalies out of the plus or minus two standard deviations, 4 are above that range, 4 are below that range. C and G also show the wind field. Blue (red) arrows indicate southward (northward) winds> Regions of the southward winds are delimited using red contours.

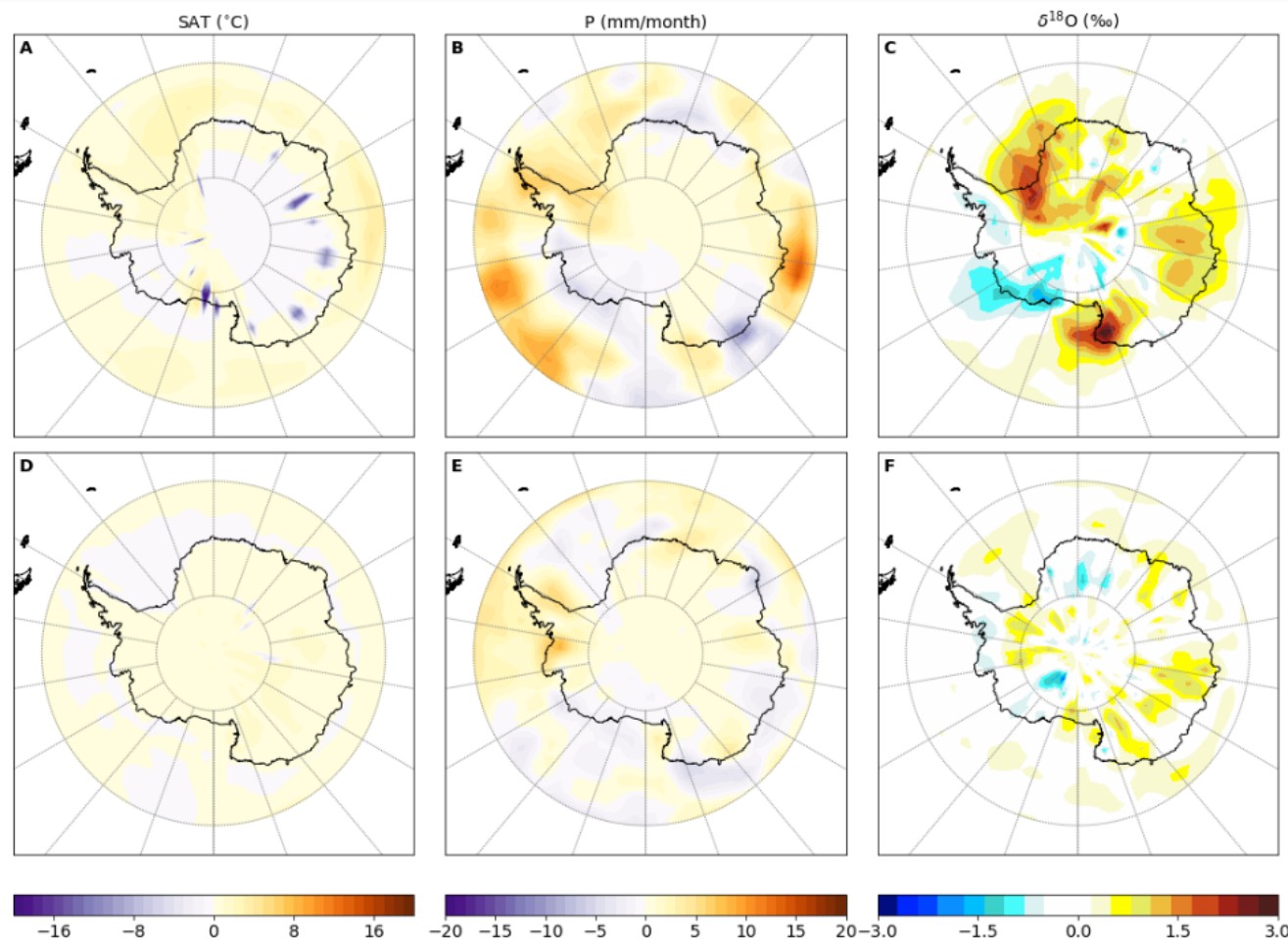

**Figure 5. Seasonal Warm-Cold changes.** Differences between the warm and cold ensemble means for the winter season (i.e. June to August, **first row: A,B,C**) and the summer season (i.e. from December to February, **second row: D,E,F**), of the surface air temperature (**A,D**, "SAT" in °C), the precipitation (**B,F**, "P" in mm/month), and the precipitated $\delta^{18}O$ (**C,F**, "$\delta^{18}O$" in ‰).

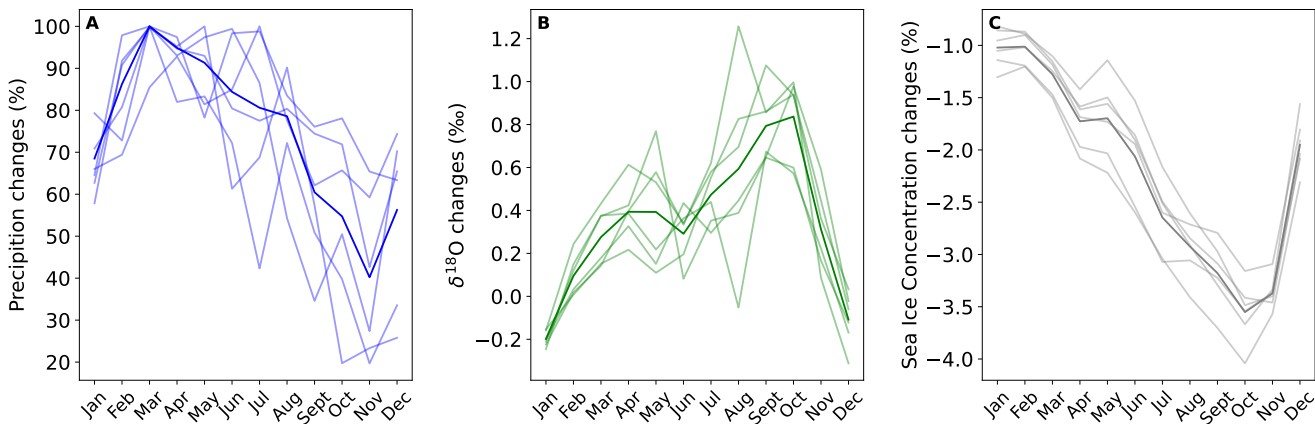

**Figure 6. Impact of climate seasonal changes.** Seasonal differences of the precipitation (in %), the precipitated $\delta^{18}O$ (in ‰) and the sea ice concentration (in %) between the first fifty simulated years and the last fifty simulated years. Light lines correspond to the member simulations and the dark line correspond to the ensemble mean.

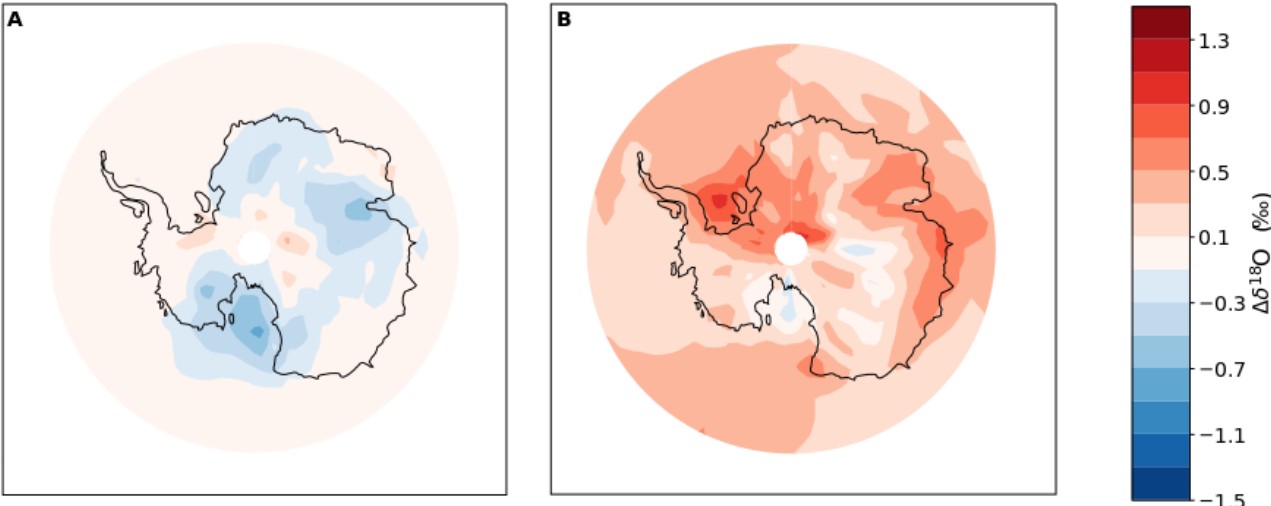

**Figure 7. The impact of seasonal changes in the precipitation and on mean annual $\delta^{18}$O on $\delta^{18}$O.** $\delta^{18}$O changes ($\Delta\delta^{18}$O in ‰) due to changes in the seasonal cycle of precipitation (A) and due to changes in the seasonal cycle $\delta^{18}$O (B) between the last fifty simulated years and the first fifty simulated years of the HadCM3 historical mean ensemble. See Methods for details of the decomposition.

## Appendix A: HadCM3 evaluation of Antarctic surface climate and $\delta^{18}O$

### A1 Method

Here, we check that HadCM3 provides a reasonable representation of the Antarctic surface climate and $\delta^{18}O$.

Surface Air Temperature (SAT) output data from HadCM3 are evaluated against the AntAWS dataset Wang et al. (2022); a compilation of Antarctic observations from 267 AWS (automatic weather station) operational between some parts of the period from 1980 to 2021. Surface mass balance (SMB) model output, calculated within the model code as precipitation minus evaporation (wind related processes are not accounted for by HadCM3), similarly are evaluated against AnSMB Wang et al.

(2021); the most recent quality-controlled published SMB compilation extracted from stakes, snow pits, ice cores, ultrasonic sounders and ground-penetrating radar. Finally, simulated $\delta^{18}O$ values are evaluated using the updated database compiled by GGoursaud et al. (2018); this combines all available firn, ice core, surface snow and precipitation observations of Antarctic $\delta^{18}O$.

We show maps and scatter plots (model versus observed values) for SAT, SMB and $\delta^{18}O$. The comparison helps establish
if the model underestimates the real spatial heterogeneity across Antarctica. Mean climatological values (20 year averages or more, averaged over the ensemble wherever possible) were calculated at each model grid point, and directly compared to the most equivalent observational climatological value (see paragraph above). The comparison uses output from a closest grid point comparison method.

### A2 Results

#### A2.1 SAT

Turner et al. (2006)'s evaluation of HadCM3 Antarctic climate, including especially near-surface air temperatures, mean sea level pressures and geopotential heights, shows a large warm bias in the Antarctic interior associated with a low-biased modeled orographic height (the heighest model gridpoint elevations do not reach 4000 m asl). This finding remains fully consistent with the newer Wang et al. (2021) observation datasets (Figure A1). The minimum climatological Antarctic plateau SAT value
is -37.2 °C (Figure A1A), considerably warmer than the AntAWS minima of -64.6 °C (Figure A1C). In regions where the observational temperature is above -30 °C the model values of SAT match the observations better, although there remains a slightly underestimating (warm bias) in West Antarctica (Figure A1B and top right of Figure A1C). Altogether, although the warm bias in the Antarctic interior contributes to weaken the linear regression between the HadCM3 simulations and the observations (correlation coefficient of 0.76), Antarctic-mean simulated SAT is surprisingly good: Antarctic-mean climatological
SAT is -25.1±14.1 and -25.0±9.1 in the observations and the HadCM3 model, respectively.

### A3 SMB

Consistent with previous studies, SMB is slightly too low in the Antarctic interior in HadCM3 Turner et al. (2006), suggesting that the warm bias in these regions do not affect the modelled SMB. The largest model SMB errors (dry and wet biases)

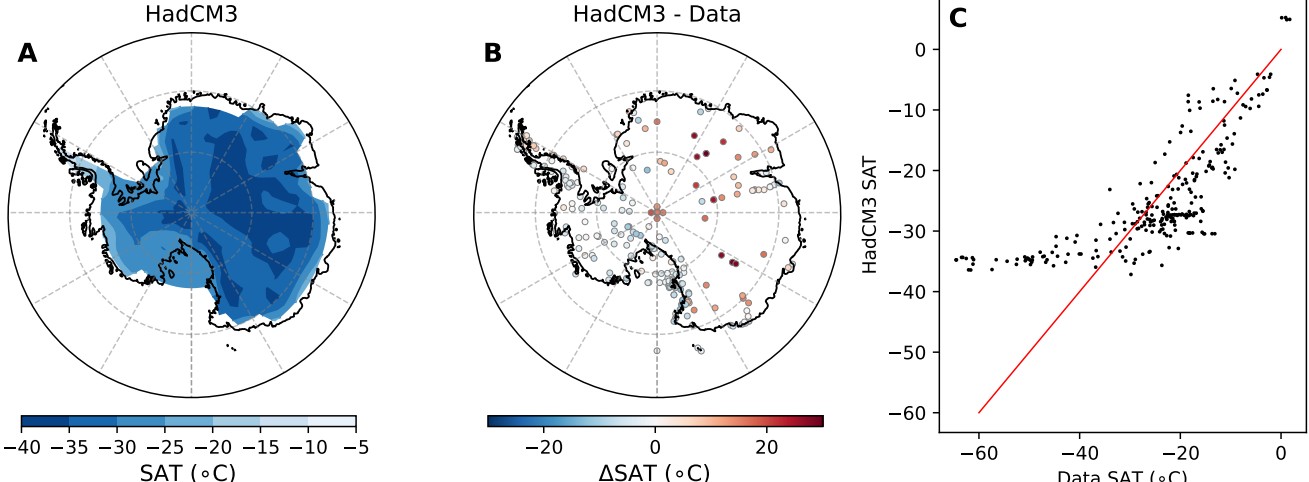

**Figure A1.** Surface Air Temperature evaluation (SAT): (A) map of the time-averaged HadCM3 SAT distribution over Antarctic resulting from the ensemble mean for the Historical period (in °C); (B) SAT difference between the time-averaged HadCM3 outputs from the ensemble mean for the Historical period, and the corresponding SAT observations (in °C); and (C) linear regression between the time-averaged HadCM3 outputs from the ensemble mean for the Historical period, and the corresponding SAT observations (black points). The red line is a 1:1 data-model slope.

occur near the coasts (Figure A2B). The dry biases may be due to the coarse HadCM3 grid, altering a realistic orography and representation of the ascending air masses that provide precipitation to these coastal regions. The coarse model grid biases can be seen on Figure A2C as a step representation of the black points compared to an expected linear regression. (Turner et al., 2006) also attribute the wet coastal biases to an overly intense mean sea level pressure field gradient: stronger than observed air flows produce excess precipitation on the west side of the Antarctic Peninsula. These aspects reduce the linear regression correlation (correlation coefficient of 0.70). The Antarctic-mean climatological SMB difference between the observations and HadCM3 is -29.7 mm/month.

## A4 $\delta^{18}$O

The distribution of the simulated $\delta^{18}$O over Antarctica is similar to observations (Antarctic-means of -36.2±9.7 ‰ and -37.4±10.3 ‰ in the observations and the HadCM3 simulations respectively; minimum values of -61.3 ‰ and -57.9 ‰ in the observations and the HadCM3 simulations respectively; maximum values of -3.2 ‰ and -7.7 ‰ in the observations and the HadCM3 simulations respectively). Excessively depleted values occur in the Antarctic interior (Figure A3). These are associated with the warm bias. Overly enriched values are observed over the Peninsula and the Weddell Sea coast, consistently with the wet bias in these region. Nevertheless, the HadCM3 historical simulations do capture the $\delta^{18}$O observations relatively

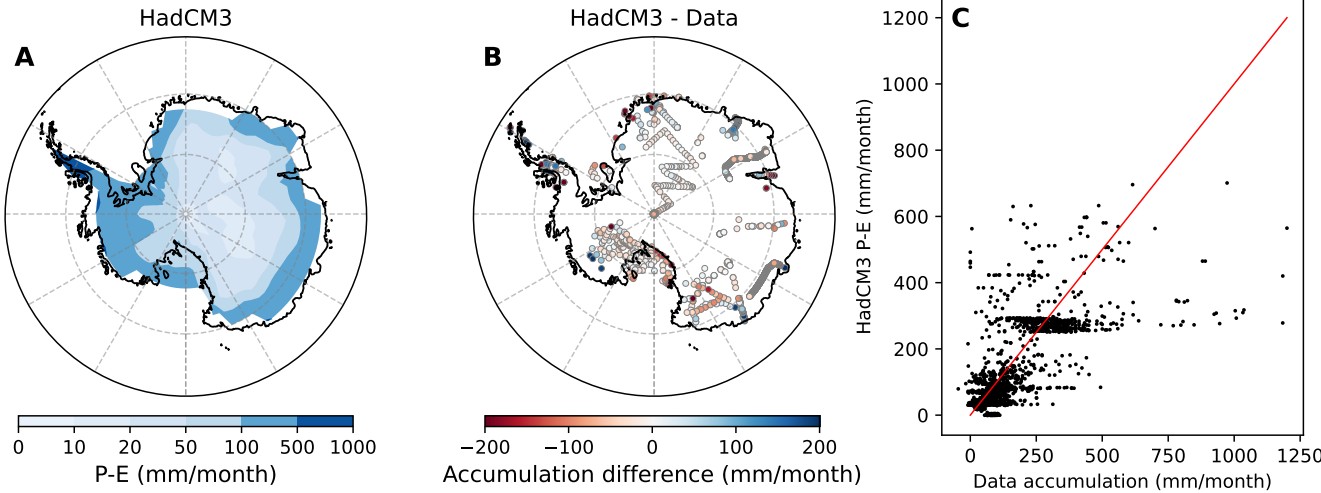

**Figure A2.** Surface Mass Balance evaluation (SMB): (A) map of the time-averaged HadCM3 Precipitation minus Evaporation (P-E) distribution over Antarctic resulting from the ensemble mean for the Historical period (in mm/month); (B) SMB difference between the time-averaged HadCM3 outputs from the ensemble mean for the Historical period, and the corresponding observations (in mm/month); and (C) linear regression between the time-averaged HadCM3 outputs from the ensemble mean for the Historical period, and the corresponding SMB observations (black points). The red line is a 1:1 data-model slope.

well, as shown by the strong relationship between the outputs and the observations (correlation coefficient of 0.84 and slope of 0.90±0.02 ‰.‰

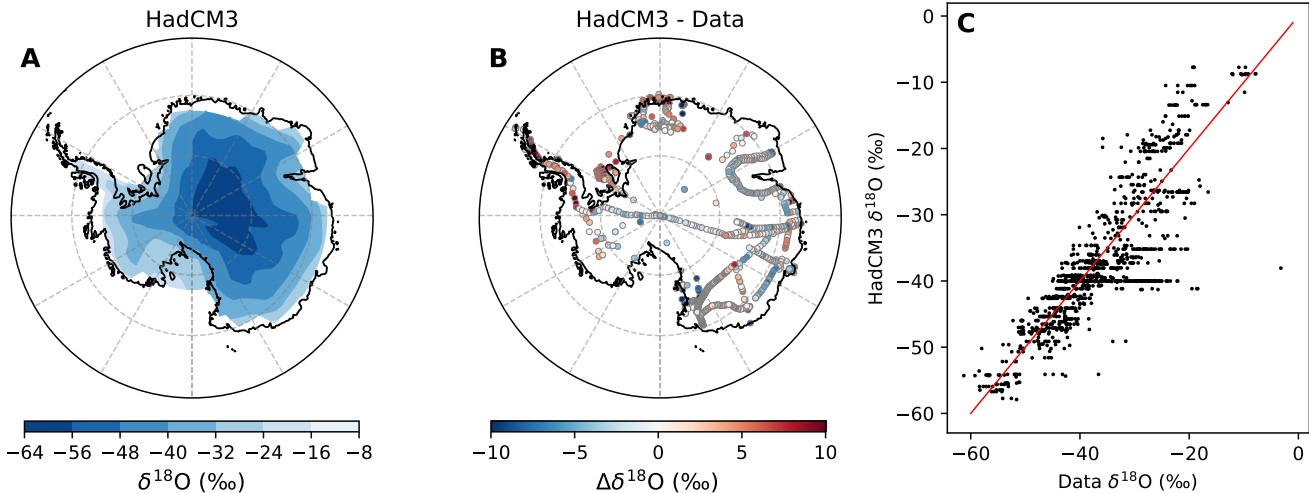

**Figure A3.** $\delta^{18}$O evaluation: (A) map of the time-averaged HadCM3 $\delta^{18}$O distribution over Antarctic resulting from the ensemble mean for the Historical period (in ‰); (B) SMB difference between the time-averaged HadCM3 outputs from the ensemble mean for the Historical period, and the corresponding observations (in ‰); and (C) linear regression between the time-averaged HadCM3 outputs from the ensemble mean for the Historical period, and the corresponding $\delta^{18}$O observations (black points). The red line is a 1:1 data-model slope.

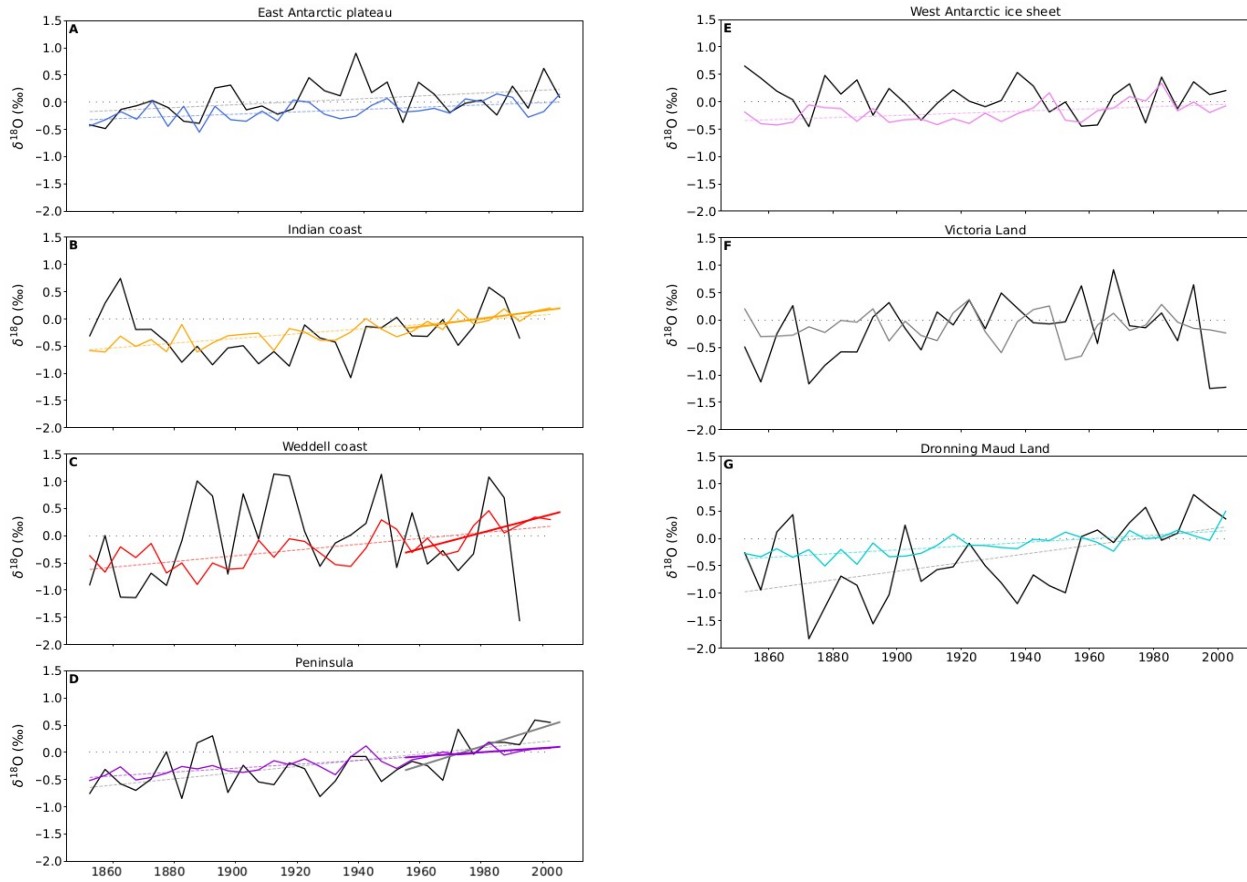

**Figure B1. Historical $\delta^{18}$O time series based on 5-years bins from ice core data and HadCM3.** Time series of Antarctic surface weighted averages of $\delta^{18}$O (in ‰) based on 5-year bin averages from ice core data (Stenni et al., 2017, black solid lines) and simulated by the HadCM3 model (ensemble mean, colored lines) for each region of Antarctica as defined in the A2k project over the historical period 1851-2004. Dashed lines correspond to linear regressions.

**Table B1.** Regional surface air temperature trends (in °C/100y): A2k reconstructions over the last 100 years (Stenni et al., 2017) based on the ECHAM5-wiso model and scaled on the climate field reconstruction from (Nicolas and Bromwich, 2014) ('A2k lower bound' and 'A2k upper bound'), as well as the HadCM3 simulated trends over the whole historical period (1851–2004). The relationships are significant.

| Region | A2k lower bound | A2k upper bound | HadCM3 |
|---|---|---|---|
| Plateau | -1.28 | -0.49 | 0.4 |
| Indian coast | 0.47 | 1.7 | 0.98 |
| Weddell coast | -0.79 | -0.5 | 0.8 |
| WAIS | 0.97 | 1.33 | 0.8 |
| Victoria Land | -0.64 | -0.55 | 0.6 |
| Dronning Maud Land | 0.98 | 1.33 | 1.11 |

## Appendix C: Regional sea ice trends

**Table C1.** Regional sea ice area trends : slope plus or minus the standard error of the slope (in $10^3$ km$^2$/y ). The correlation coefficients are given into brackets. Only significant relationships are given. The sea ice regions are defined in term of longitudes as follows: the Indian sector is limited between 20 °E and 90 °E. The Pacific sector is limited between 90 °E and 160 °E. The Ross sector is limited between 160 °E abd 230 °E. The Bellingshausen – Amundsen is limited between 230 °E and 300 °E. Finally the Weddell sector is limited between 300 °E and 20 °E.

| Region | Historical | Last 50 years |
|---|---|---|
| Indian | -4.9±0.5 (-0.67) | -5.5±2.3 (-0.32) |
| Pacific | -2.5±0.4 (-0.44) | -4.9±2.0 (-0.33) |
| Ross | -4.0±0.4 (-0.60) | -8.5±2.0 (-0.33) |
| Bellingshausen Amundsen | -3.5±0.5 (-0.51) | -7.9±2.5 (-0.41) |
| Weddell | -2.4±2.4 (-0.64) | – |

## Appendix D: Dependency of the time window on the HadCM3 $\delta^{18}$O trends

**Table D1.** Antarctic regional $\delta^{18}$O trends ("Plateau", "Indian Coast", "Weddell Coast", "Weddell Coast", "Peninsula", "WAIS", "Victoria Land", and "Dronning Maud Land", with region delimitations as defined in (Stenni et al., 2017)) simulated by the HadCM3 model using our Historical mean ensemble using different time window lengths: the 153, 100, 75, 60, 45, and 30 last simulated years. The gradient and the standard error of the gradient are given for each linear relationship, associated with the correlation coefficient into brackets. Non significant relationships are indicated by a dash.

| Window length (years) | 153 | 100 | 75 | 60 | 45 | 30 |
|---|---|---|---|---|---|---|
| Plateau | 0.02±0.01 [0.19] | - | - | - | - | - |
| Indian coast | 0.04±0.01 [0.45] | 0.05±0.01 [0.36] | 0.07±0.02 [0.4] | 0.06±0.03 [0.31] | - | - |
| Weddell coast | 0.05±0.01 [0.39] | 0.06±0.02 [0.29] | 0.09±0.03 [0.35] | - | 0.17±0.06 [0.4] | - |
| Peninsula | 0.04±0. [0.57] | 0.04±0.01 [0.42] | 0.04±0.01 [0.35] | 0.04±0.02 [0.27] | - | - |
| WAIS | 0.02±0.01 [0.24] | 0.04±0.01 [0.32] | - | - | - | - |
| Victoria Land | - | - | - | - | - | - |
| Dronning Maud Land | 0.03±0.01 [0.45] | 0.04±0.01 [0.33] | 0.05±0.01 [0.33] | 0.05±0.02 [0.33] | 0.1±0.04 [0.4] | - |

# Appendix E:  Statistical description of the HadCM3 cold and warm ensembles

**Table E1.** Statistical description of the surface air temperature ("SAT", in $^\circ$C), precipitation ("Precip", in mm/month) and $\delta^{18}$O (in ‰) for the cold and warm ensembles: minimum ("min"), maximum ("max"), mean $\pm$ the standard deviation ("mean$\pm$std").

| Variable | Cold ensemble | | | Warm ensemble | | |
|---|---|---|---|---|---|---|
| | Min | Max | Mean$\pm$Std | Min | Max | Mean$\pm$Std |
| SAT | -1.29 | 0.85 | -0.49$\pm$0.36 | -0.92 | 1.39 | 0.47$\pm$0.34 |
| Precip | -4.0 | 3.4 | -0.71$\pm$0.77 | -3.1 | 5.8 | 0.54$\pm$0.60 |
| $\delta^{18}$O | -1.94 | 0.28 | -0.62$\pm$0.42 | -0.07 | 1.6 | 0.65$\pm$0.33 |

# Appendix F: $\delta^{18}$O – SAT gradients at ice core locations simulated by HadCM3

**Table F1.** $\delta^{18}$O – SAT gradients in ‰/°C at ice core locations. Gradients are accompanied with the standard error. Correlation coefficients are given into brackets. Finally, numbers in italic correpond to non significant relationships (p>0.05).

| | |
|---|---|
| Vostok | *0.04±0.04 [0.07]* |
| Dome F | **0.11±0.05 [0.16]** |
| EDC | **0.23±0.07 [0.26]** |
| EDML | *0.01±0.03 [0.04]* |
| Talos | **-0.09±0.04 [-0.18]** |
| Taylor Dome | *0.06±0.07 [0.08]* |
| WDC | **0.10±0.04 [0.2]** |
| Dome B | *0.07±0.04 [0.13]* |
| Law Dome | **0.65±0.1 [0.46]** |
| Siple Dome | *0.08±0.06 [0.12]* |
| Byrd | *0.07±0.05 [0.12]* |
| Dome A | *0.04±0.04 [0.12]* |
| RICE | **0.07±0.03 [0.18]** |
| Fletcher | **0.21±0.03 [0.54]** |
| James Ross | **0.25±0.03 [0.58]** |
| Berkner | **0.49±0.05 [0.64]** |
| Skytrain | **0.43±0.08 [0.41]** |
| Hercules Dome | *0.003±0.03 [0.01]* |

**Appendix G: Impact of the Southern Annular Mode**

The Southern Annular Mode (SAM) is the leading mode of atmospheric variability in the Southern Hemisphere (Thompson and Wallace, 2000). Especially, it describes the position and the strength of the polar jet position, the southern westerly belt and the associated storm tracks. A positive (negative) phase of the SAM is associated with an intensified (weakened) poleward (northward) shift of the polar jet. The SAM is thus the preferred studied mode to investigate the Southern Hemisphere teleconnection with lower latitudes.

Here, we used the definition of the SAM index following the approach of Gong and Wang (1999), as the difference between the normalized monthly zonal mean sea level pressure between 40°S and 65°S. Here we used the period 1961–1990 as a reference interval.

$$SAM = \frac{P_{40} - \mu_{40}}{\sigma_{40}} - \frac{P_{65} - \mu_{65}}{\sigma_{65}}$$

where $P_{40}$ and $P_{65}$ are the monthly mean sea level pressure at 40°S and 65°S, $\mu_{40}$ and $\mu_{65}$ are the mean of the monthly mean sea level pressure at 40°S and 65°S over the reference interval 1961–1990, and $\sigma_{40}$ and $\sigma_{65}$ are the standard deviations of the monthly mean sea level pressure at 40°S and 65°S over the reference interval 1961–1990.

We computed the linear regressions between the calculated SAM and our climate variables (Figure G1): (i) the surface air temperature (SAT), (ii) the precipitation (P) and finally (iii) the precipitation weighted $\delta^{18}O$ ($\delta^{18}O$). These linear regressions were computed over the whole Historical simulated period, as well as for the recent period 1950–2004, at the annual scale. Note that, as done in the main corpus of the manuscript, we computed these linear regressions using the stack of the ensemble members, resulting in 918 points for the Historical period (1851–2004) and 324 points for the period 1950–2004.

Within the frame of the CMIP5 project, the ability of HadCM3 to reproduce the SAM was evaluated (Zheng et al., 2013). As for all the CMIP5 models, HadCM3 overestimates the SAM index variability (Zheng et al., 2013; Zhang et al., 2022). Nevertheless, it reproduces the decadal variability of the SAM index and displays the best correlation coefficient between modeled and observed detrended SAM index (Zheng et al., 2013).

Previous studies reported, based on observations, that main of the Antarctic continent is globally colder and drier while the SAM is in a positive phase, as the stronger southern westerly wind belt reduces the exchanges with warmer air masses from midlatitude regions, at the exception of the Peninsula (Clem et al., 2016). These effects are reproduced in our HadCM3 simulations, as shown by the correlation coefficient values between the SAM and the SAT that are positive over the northern Antarctic peninsula, but negative over the rest of the continent, especially on coastal areas (Figures G1A and G1D).

Similarly, it was shown that there is less southward moisture advection towards the Antarctic interior in a positive phase of the SAM, reducing precipitations. In our simulations (Figures G1B and G1E), this effect is enhanced over the Antarctic plateau, Victoria Land and Marie Byrd Land. At the opposite, the Antarctic peninsula receives more precipitation. However, the discrepancy in the HadCM3 orography unables the «shadow effect» decreasing precipitation on the Eastern part of the peninsula due to the presence of mountains (Fogt and Marshall, 2020).

The link between water stable isotopes and the SAM is less settled. A couple of publications displayed a correlation between the water stable isotope content in ice cores and the SAM index, but no systematic method allowed an established link. For

instance, (Servettaz et al., 2022) suggest some impacts of the SAM on the isotopic content of the Aurora Basin North ice core over the last millennium, although not on the whole length of the core. Also, over the Fimbull Ice Sheet, Vega et al. (2016) suggest that the absence of correspondence between water stable isotopes and SAT might be explained by changes in atmospheric circulation, supported by a high correlation between d-excess measured in the KM and BI ices cores and the SAM index. Kino et al. (2021) showed the contribution of SAM over precipitation weighted $\delta^{18}O$ at the daily scale simulated by the MIROC5-iso model nudged toward the JRA-25 reanalyses, over the period 1981–2010 at Dome Fuji. However, they warn that it does no prevail on all antarctic locations of the Antarctic plateau. For instance, Dome C is less sensitive to SAM compared to possible other teleconnections modes (Dreossi et al., 2023). In our simulations, the correlation coefficients between the SAM and precipitation weighted $\delta^{18}O$ are significant and negative over the whole continent (Figures G1.C and G1.F), but remain week, with a mean of -0.26±0.11 over the Historical period and -0.27±0.12 for the period 1950–2004. From our simulations, we thus cannot neither establish a robust link between the SAM and the Antarctic precipitation weighted $\delta^{18}O$.

However, studying the impact of the atmospheric circulation change on Antarctic precipitation weighted $\delta^{18}O$ should not be boiled downed to the link with the SAM. For instance, only some El Nino Southern Oscilation (ENSO)/SAM combinations (El Nino/negative SAM and La Nina/positive SAM) contribute to strenghen the Amundsen Sea Low (e.g. Wilson et al., 2016), as observed through the analysis of the the Roosevelt Island Climate Evolution (RICE) $\delta^{18}O$ Emanuelsson et al. (2023). SAM-induced processes impacting Antarctic precipitation weighted $\delta^{18}O$ are also not trivial: SAM changes SAT, precipitation regimes but also the sea ice in a more complex manner (Fogt and Marshall, 2020). Other modes affect the Antarctic atmospheric circulation and might explain the $\delta^{18}O$ changes, as for the Indian Ocean Dipole in phase with El Nino through the production of atmospheric rivers (Shields et al., 2022).

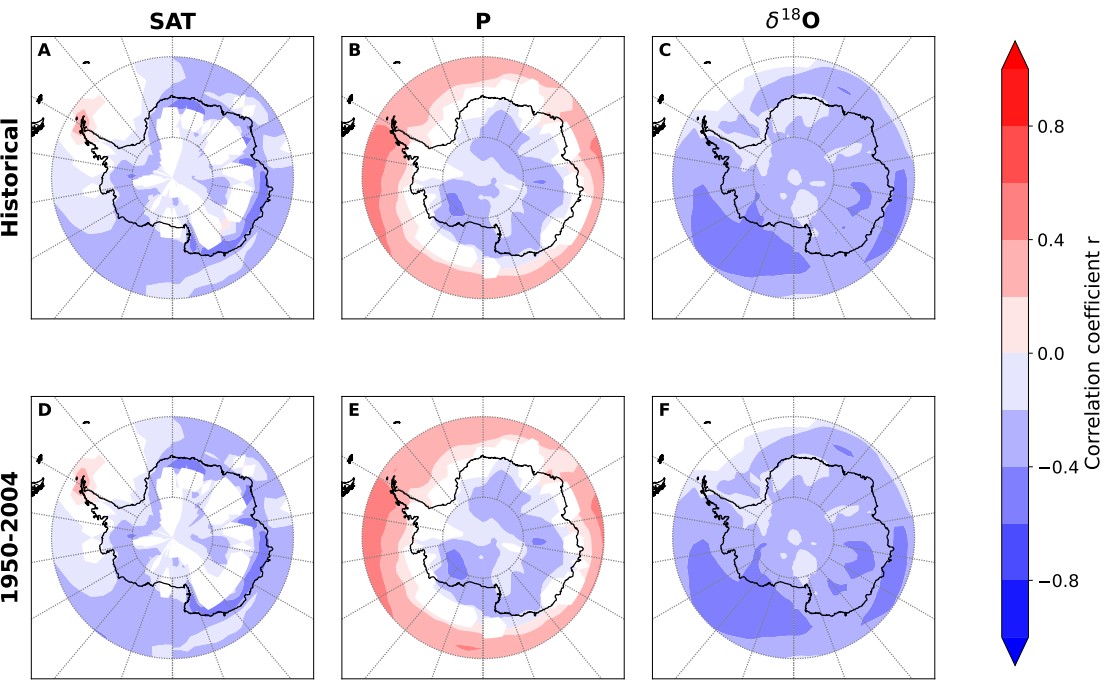

**Figure G1.** Correlation coefficients between the Southern Annular Mode index and the Surface Air Temperature ("SAT", A and D), the precipitations («P», B and E), and the precipitation weighted $\delta^{18}O$ (C and E) simulated by the HadCM3 model at the annual scale for the Historical Period (1851–2004, first row) and the 1950–2004 period (second row). Only significant relationships are shown (p-value<0.05).