# Peer review of "Decoupling of $\delta^{18}$ O from surface temperature in Antarctica in an ensemble of Historical simulations"

_EGUsphere, 2023_

## Referee Comment (RC2)

**Review of Decoupling of δ18O from surface temperature in Antarctica in an ensemble of Historical simulations**

The manuscript employs simulations generated by an isotope-enabled General Circulation Model (GCM) to assess the temporal variations in isotopic composition, surface temperature, precipitation, and sea ice concentration within Antarctica over the past 200 years. Specifically, they suggest that differences between the simulated isotopic composition and temperature variations can be attributed to change of precipitation patterns due to the impact of sea ice concentration on the moisture pathways toward Antarctica.

The topic addressed by the manuscript is extremely important because the accuracy of the isotopic paleothermometer is directly affected by the link between isotopic composition in ice cores and temperature, which is only constrained empirically. Models are invaluable tools to study and explore the relationship between isotopic composition and temperature spatially, temporally, and with the different time scales. Here, the study based on the outputs of a single model (with occasional comparison with results from other studies using another one) lacks robustness to provide concrete evidence that can help strengthen our understanding of the isotopic paleothermometer. While the study suggested by the authors is worth pursuing, several shortcomings weaken what could otherwise be an important study for the field.

**General comments:**

The manuscript predominantly relies on the outcomes of multiple runs from the HadCM3 model. Although the authors note in the Method section that 'HadCM3 provides a reasonable representation of Antarctic Climate and δ18O,' there is a notable absence of a critical evaluation of the model's performance in comparison to available observations in Antarctica. The manuscript does not provide values indicating potential biases or errors in the model. It is challenging to assess the trustworthiness of the model outputs without a direct comparison with present-day observational data in HadCM3 iso, similar to the approach in Figure 1 of (Werner et al., 2018). This becomes more significant as the manuscript critiques observations without addressing potential biases in the studied model, even though the IPCC specifically relies on a diverse ensemble of models to balance potential biases. While a few articles are referenced (specifically (Holloway et al., 2016; Tindall et al., 2009)) that offer some comparisons with discrete values, the absence of confidence intervals makes it difficult to evaluate the robustness of the relationship between isotopic composition and temperature. This limitation affects the confidence in the authors' results, particularly given the extensive literature containing data that could be directly compared with the model outputs for temperature data (Jones et al., 2018)(Jones et al., 2018), isotopic composition (Dittmann et al., 2016; Ekaykin et al., 2002; Landais et al., 2017; Masson-Delmotte et al., 2008; Schlosser et al., 2004; Stenni et al., 2016; Touzeau et al., 2016), NSIDC products for the sea ice…

Some of the results are compared with outputs from another General Circulation Model (GCM), specifically ECHAM5wiso, to assess 'model dependency.' This comparison reveals substantial differences, with predicted slopes between isotopic composition and temperature regionally or across all of Antarctica showing almost a 100% disparity. These findings contribute to a diminishing confidence in the outputs of the HadCM3 model. It's worth noting that the comparison involves an older version of ECHAM with isotopes (ECHAM6wiso, released in 2021, has been demonstrated to be more accurate according to (Cauquoin and Werner, 2021)). Additionally, ECHAM5wiso was nudged with ERA40, a choice that may seem unconventional given the availability of more recent products like ERA-interim in 2006 (Dee et al., 2011) and ERA5 in 2018 (Hersbach et al., 2020).

Overall, the manuscript does not engage in a critical discussion regarding the capabilities of GCMs (CMIP5 or CMIP6) to accurately depict regional-scale climate dynamics. This is noteworthy given the existing body of literature highlighting that the simulated variability often underestimates (Laepple and Huybers, 2014; Shao and Ditlevsen, 2016; Zhu et al., 2019). It would be beneficial for the manuscript to address the impact of discrepancies between modelling outputs and observations within this context. A thoughtful consideration of this aspect would enhance the argument and bolster confidence in the assertion that the results from HadCM3 are reliable in this context.

The comparison with observational products seems to selectively draw from publications that align with the authors' findings, potentially overlooking contributions that present contrasting results (Clem et al., 2020; Jones et al., 2019) or dismissing them without explicit justification ((Casado et al., 2023), in which I was involved as a lead author). There are several instances detailed below in the specific comments when the authors obtained different results using the same calculation (trends from linear regression) and presumably the same datasets (PAGES 2k) and obtain different results without providing an explanation. Notably, in (Casado et al., 2023), considerable effort was invested in validating trend calculations using the products from (Stenni et al., 2017). It would be valuable for the manuscript to articulate why different trends are identified here, especially as the current narrative suggests a potential error in (Casado et al., 2023). While the scientific method encourages revisiting and improving upon previous results, the manuscript should provide evidence-based arguments when challenging peer-reviewed scientific findings, moving beyond statements like 'It is unclear why the trend in (Casado et al., 2023) is higher."

In Section 5 (*Drivers of $\delta^{18}O$ changes*), there appears to be a lack of evidence supporting the hypothesis regarding the mechanisms behind $\delta^{18}O$ variations. It is commonly acknowledged that correlation does not necessarily imply causation. In this context, the connections between $\delta^{18}O$ and other climatic parameters are not based on correlation but on the representation of end-members (warmest and coldest conditions) on a map. The linkage between $\delta^{18}O$ and temperature, specifically attributing the decoupling to sea ice concentration as depicted in Figures 4 and 5, is unclear. Notably, the authors seem to have overlooked the potential impact of the SAM, which has been suggested to influence both isotopic composition (Kino et al., 2021) and sea ice concentration around Antarctica (Eayrs et al., 2021).

**Specific comments:**

Lines 5 to 6: "Our ensemble captures observed historical SAT and precipitation trends, and weak δ18O trends."
Currently, this statement is not supported within this manuscript. It does conflict with recent publications on the topic (Casado et al., 2023; Clem et al., 2020; Jones et al., 2019) without clear justifications for considering this manuscript's results as more valid than those published in peer-reviewed journals

Lines 6 to 7: "The weak δ18O trends mean there is no significant relationship between SAT and δ18O over one third of Antarctica, and also half of our considered ice core sites, though relationships are stronger when using regional averages."
While it can be debated, it's important to note that in Antarctica, every site where precipitation isotopic composition has been sampled demonstrates a remarkably robust correlation with temperature (Dittmann et al., 2016; Fujita and Abe, 2006; Landais et al., 2012; Schlosser et al., 2004; Stenni et al., 2016; Touzeau et al., 2016). Any counterargument should be accompanied by evidence explaining why the findings in this manuscript should be considered more valid than those published in peer-reviewed journals.

Lines 19 to 21: "The collapse of Antarctic ice shelves have similarly increased in frequency (Graham et al., 2022; Milillo et al., 2022; Wille et al., 2022), with a 1,600 square kilometers iceberg breaking away from the Brunt ice shelf on January 22nd, 2023."
This seems anecdotal and unrelated to the topic of the manuscript.

Lines 26 to 27: "The remote nature of this vast continent means that observational data covering Antarctica are sparse in both space and time (Turner et al., 2004)."
Rewrite, too vague and imprecise. Satellite data are observational, and they are regularly gridded and relatively dense. Also, most of the observations are in the last 50 years, so they are not "sparse in time", but rather extremely concentrated on a specific period. I understand what you meant, but the sentence could be improved.

Lines 27 to 28: "The reconstruction of past temperature change is thus of paramount importance for understanding natural variability, versus effects in response to anthropogenic climate change"

"paramount importance" does not seem appropriate here. "versus effects in response…" is nonsensical.

Lines 34 to 35: "it is sometimes referred as the 'ice core paleothermometer' (Lorius and Merlivat, 1977; Masson et al., 2000)."
It is referred as the "Isotopic paleothermometer" in the specific references that are cited here. Also Lorius and Merlivat 1977 does not qualify as a peer-reviewed publication, consider replacing by (Lorius et al., 1969). Other proxies from ice cores can be used for past temperature reconstructions, for instance the borehole thermometry.

Line 36: "The paleothermometer" is not sufficient, should be describe as "isotopic paleothermometer".

Lines 38 to 41: "This ice core based record was then used to show that simulated temperatures from the Atmospheric General Circulation (AGCM) models run in the frame of the Coupled Model Intercomparison Project Phases 5 (Taylor et al., 2012) and 6 (Eyring et al., 2016, CMIP6) are too low. However, the paleothermometer relationship has been shown to vary spatially over the Antarctic continent (e.g. Sime et al., 2008, 2009a)."
(Casado et al., 2023) does not state that the temperatures are not too low, but that the variability (natural and forced) is too low. In addition, variable isotope-temperature conversions are used for the different regions of Antarctica in Casado et al, 2023, so it seems that the use of "however" here suggests an opposition that is not based on actual opposite point of views.

Line 49: "The geographical variability in the 'paleothermometer' is due to controls on d18O other than SAT."
This statement appears slightly misleading in the sense that it doesn't reflect that Rayleigh distillation (Dansgaard et al, 1964) suggests already that the signal acquired across the distillation pathway, and not a pure local surface temperature signal.

Lines 59 to 62: "AGCM isotopic studies have focused on the effects of external forcing on the SAT–δ18O relationship, including elevation and greenhouse gases across a range of timescales (e.g. Sime et al., 2009b; Werner et al., 2018; Goursaud et al., 2021). A major result is that, for differing time-scales and driving mechanisms, different SAT–δ18O relationships can be obtained."
This sentence suggests that this result was only found using isotope enabled GCM studies, while several proxy based studies have also shown this, including relatively old ones, see for instance (Guillevic et al., 2013; Jouzel, 1999).

Lines 78 to 83: "The Historical simulation protocol was defined in the frame of the CMIP Phase 6 (Eyring et al., 2016), with an express purpose to investigate the anthropogenic forcing on climate (Johns et al., 2003) and serve as a benchmark to evaluate model performance (Andrews et al., 2020; Miller et al., 2021; Parsons et al., 2020; Rong et al., 2021; Roach et al., 2020). There have been few examples of studies using Historical simulations focused over Antarctica (Gao et al., 2021; Purich and England, 2021; Raphael et al., 2020; Roach et al., 2020).
Here, we use the Hadley Center general circulation model (HadCM3; GCM), to run six transient Historical simulations."
This is technically correct, it also implies that HadCM3 is a CMIP6 model, although it is actually a CMIP5 model. The description should first be the model. Then be the transient historical simulations that were done. At this point, for non-modeller specialist such as myself, it is not possible to know what the *historical simulation protocol* entails, so rather than giving examples of studies, the key important points that a reader should know should be described here.
For instance, it is not clear if/how the surface conditions were prescribed since only the atmospheric component was used. Was the sea ice concentration simulated or prescribed? If it was prescribed, since HadGEM performs relatively poorly for sea ice concentration, surface ocean temperature, and to some extent for the 850hPa temperature (Agosta et al., 2015), how is that affecting the results?

Lines 89 to 90: "HadCM3 provides a reasonable representation of Antarctic climate and δ18O (Turner et al., 2006; Tindall et al., 2009; Holloway et al., 2016)."

Tindall et al, 2009 provides a comparison of d18O with observations from mostly tropical and temperate regions, with only 2 data points in Antarctica. Holloway et al, 2016 provides a comparison of the outputs of the model and 4 ice core records during the last glacial maximum. To my knowledge, no comparison between modern $\delta^{18}O$ from observations and model outputs has been published. Considering the large warm bias that most of the isotope enabled CMIP5 models suffered which have been fixed in CMIP6 versions (Cauquoin and Werner, 2021; Werner et al., 2018), it feels like this statement does not provide the necessary information to know if we can trust the outputs during the historical periods. Please consider reproducing Figure 1 of (Werner et al., 2018).

Lines 96 to 97: "Where we regress climate variables against δ18O, the linear regressions are computed using the stacked individual ensemble members, rather than using the ensemble mean."

Unclear to me. Are you computing the linear regression on a stack of all the individual members ? Or are you computing it against individual members and then stacking the linear regression ? The former seems fairly similar than using the ensemble mean.

Lines 99 to 101: "Our Historical SAT–d18O linear relationship at the regional scale, as well as at the nearest model grid-cell to each ice core location, are compared with the ECHAM5-wiso slopes and correlation coefficients provided in Stenni et al. (2017)."

Why would you use ECHAM5-wiso here when your manuscript is about HadCM3 ? If you're using an isotope enabled version of ECHAM, why not use ECHAM6 which has been released in 2021. And if the goal is to provide a reference that is published, why is the comparison with observations not included, i.e. (Casado et al., 2017; Fujita and Abe, 2006; Masson-Delmotte et al., 2008; Schlosser et al., 2004; Stenni et al., 2016; Touzeau et al., 2016).

Line 114: "3 Trends in Antarctic SAT, precipitation, sea ice and δ18O"

Is this a result section ? it seems to include results and discussion, but then section 4 and 5 as well. While I don't think Climate of the Past has a strict rule on which structure to use for manuscript, I am not convinced that the classical structure wouldn't help the readability of the manuscript.

Lines 122 to 123: "This is consistent with observations of 0.12±0.07 °C per decade over 1957-2006 (Steig et al., 2009) and 0.11±0.08 °C per decade over 1959-2012 (Nicolas and Bromwich, 2014)."

This is only partially true. Jones et al, 2019 reports larger warming across Antarctica when the SAM-congruent trend has been taken into account. Clem et al, 2019 reports temperature increase of 0.6°C per decades at the south pole station, where the map in Figure 1E. reports actually a cooling between 1850-1900 and 1950-2000. ERA5 reports as well a large warming across Antarctica, which is indeed a reanalysis based on satellite observations.

Also, is the comparison really accurate if you compare on the one hand model outputs over all of Antarctica and meteorological observations from N&B which are clearly biased toward coastal regions?

Lines 129 to 130: "Forecast System Reanalysis (CFSR), and 7.1±1.5 mm/y per decade from the National Centers for Environmental Prediction reanalyses 2 (NCEP-2) over 1979-2009."

It seems that different sources of data (observations, reanalyses, or other type of models) are used for different variables. Wouldn't it be valuable to compare all of your variables with one systematic source of data, may it be direct observations, satellite observations, reanalyses…

Lines 166 to 168: "Despite the simulated increases in SAT and precipitation, d18O shows a very weak trend of 0.04 ±0.003‰per decade (r=0.21) over the last 50 years. Interestingly, (Casado et al., 2023) provide a higher trend from 1950–2005 of 0.11± 0.02‰per decade. It is unclear why the trend in (Casado et al., 2023) is higher."

The response provided seems insufficient, particularly in light of the extensive sensitivity tests

conducted in (Casado et al., 2023)to elucidate the disparities with the trends observed in S(Stenni et al., 2017). It would be helpful to have clarification on the handling of isotopic data from the PAGES2k network in this context, such as the methodology for averaging monthly isotopic data with annual and interannual data. Given that a trend is essentially a mathematical representation, and both this manuscript and Casado et al., 2023 utilise the same dataset, it raises concerns about the disparity in values. Moreover, the methods employed in Casado et al., 2023 were replicated using the outputs generated in Stenni et al., 2017, resulting in a slope of 0.10 permil per decade. Additionally, an alternative method based on dynamical system theory yielded an even larger value.

Lines 169 to 170: "Before this, we provide a brief overview of the regional picture. At the regional scale, over the Historical period, trends are small (Figure 2)."
The regions in Figure 2 seem to be different than the ones in Stenni et al 2017 beyond the impact that the model grid would do to the attribution. For instance, none of the coastal grid points are included in your analysis, which differs strongly from Stenni et al. Victoria Land region extend further east near the coast (with the strong consequence of adding an additionnal core in this grid point compared to Stenni et al). Another notable difference comes from the lack of the coast part in DML coast region, which means that almost none of the ice core available for this region are represented in your average. As this seems to be a significant difference compared to Stenni et al, 2017, where several pages were included to explain the choice of the region, it needs to be justified.

Lines 171 to 172: "and is the highest for the Weddell coast with a trend of 0.05 ‰ per decade (r=0.39), and the strongest for the peninsula with a trend of 0.04 ‰ per decade (r=0.57)."
Which one is it ? The "highest" and the "strongest" should be the same.

Lines 183 to 185: "These disparities could be explained by the different time windows, the different methodologies or the lack of ice core data to make representative regional reconstructions."
All of these hypotheses can be readily examined to ensure the robustness of the arguments detailed here. For the first hypothesis, it might be beneficial to incorporate a table in the supplementary materials, offering a comparison for the same time windows. Adaptations in methodologies can be explored, and Supplementary Table S3 in Casado et al., 2023, already presents trends using both their approach and the one from Stenni et al., 2017. Additionally, the absence of ice cores to establish representative regional reconstructions can be tested by focusing solely on specific grid points of the model corresponding to the ice core locations, comparing them to the regional average encompassing all grid points in the region. A fourth option, which the author does not explicitly address, is the potential bias or insufficient representation of variability in HadCM3, as suggested by Casado et al., 2023, particularly for most CMIP models.

Lines 194 to 206: This is an interesting discussion, but again, it fails to address the elephant in the room which is the model biases. The maps show areas with non-significant link between isotopic composition and temperature in the model, which seems sound and robust, but how does it compared to observations in the field? For instance, no correlation is found at the site of Vostok, where precipitation isotopic composition shows a significant correlation (R = 0.63, slope of 0.35 permil per degree) (Touzeau et al., 2016), while the map suggest a slope of 0 with non-significant correlation. The slope and correlation at Dome C also seems lower (below 0.3 with a r <0.3) than in observations (R² = 0.63 and slope of 0.49) (Stenni et al., 2016). In general, any discussions which could support the validity of the model outputs would strengthen the manuscript, or at least provide confidence interval on the range of values that can actually be interpreted.

Lines 211: "Here, and also for other warm climate results"
Is it actually relevant? The manuscript is about historical reconstructions. This could potentially be discussed in the end of the discussion, but this subsection feels like results.

Lines 219 to 220: "Interestingly, however, this is not the case when comparing between the last 50 years of our HadCM3 simulation and the ECHAM5-wiso simulation."
It is obsolete to compare your result with yet another CMIP5 model, when the isotope enabled version ECHAM6 wiso is available for more than 3 years.

Lines 225 to 227: "all the historical SAT-d18O relationships are different from the LGM-PI ECHAM5, and LIG-PI HadCM3 relationships: Werner et al. (2018) report LGM-PI regional gradients in ECHAM5 that are 17-26% lower, while (Sime et al., 2009b) and (Holloway et al., 2016) present LIG-PI regional gradients that are ~50% lower for HadCM3."
Nobody would expect the historical and the LGM-PI relationships to be the same, considering the difference of time scales and underlying mechanisms driving the temperature changes. Is this really necessary in this manuscript which is about historical changes in the isotope-temperature relationship?

Lines 228: "ECHAM5 towards ERA-40 reanalysis,"
There are two generations of newer ERA products. It is unclear why the authors did not use either of these products. If the nudging is not conducted with the newer products in a revised version of the manuscript, which is what is really needed here, a clear justification for this omission will be required.

Lines 238 to 239: "The primary mechanism driving continental-scale SAT-d18O decoupling is the simulated loss of sea ice during the historical period (Figure 5DH)."
The rationale that could explain how to make this assessment is not supported by Figure 4D to H. The patterns of sea ice concentration anomalies does not explain any link with the variations of temperature and isotopic composition inside Antarctica by itself. There is no correlation provided, no mechanism, no simulations in which the sea ice concentration is artificially varied to support this assessment. Overall, this entire section falls short in establishing any form of causality and warrants a comprehensive revision.

The conclusion should be revised once the rest of the manuscript has been reassessed.

**Bibliography**

Agosta, C., Fettweis, X. and Datta, R.: Evaluation of the CMIP5 models in the aim of regional modelling of the Antarctic surface mass balance, Cryosph., 9(6), 2311–2321, doi:10.5194/tc-9-2311-2015, 2015.

Casado, M., Orsi, A. J. and Landais, A.: On the limits of climate reconstruction from water stable isotopes in polar ice cores, Past Glob. Chang. Mag., 25(3), 146–147, doi:10.22498/pages.25.3.146, 2017.

Casado, M., Hébert, R., Faranda, D. and Landais, A.: The quandary of detecting the signature of climate change in Antarctica, Nat. Clim. Chang. [online] Available from: https://doi.org/10.1038/s41558-023-01791-5, 2023.

Cauquoin, A. and Werner, M.: High-Resolution Nudged Isotope Modeling With ECHAM6-Wiso: Impacts of Updated Model Physics and ERA5 Reanalysis Data, J. Adv. Model. Earth Syst., 13(11), e2021MS002532, doi:https://doi.org/10.1029/2021MS002532, 2021.

Clem, K. R., Fogt, R. L., Turner, J., Lintner, B. R., Marshall, G. J., Miller, J. R. and Renwick, J. A.: Record warming at the South Pole during the past three decades, Nat. Clim. Chang., 10(8), 762–770, 2020.

Dee, D. P., Uppala, S. M., Simmons, A. J., Berrisford, P., Poli, P., Kobayashi, S., Andrae, U., Balmaseda, M. A., Balsamo, G. and Bauer,  d P.: The ERA-Interim reanalysis: Configuration and performance of the data assimilation system, Q. J. R. Meteorol. Soc., 137(656), 553–597, 2011.

Dittmann, A., Schlosser, E., Masson-Delmotte, V., Powers, J. G., Manning, K. W., Werner, M. and Fujita, K.: Precipitation regime and stable isotopes at Dome Fuji, East Antarctica, Atmos. Chem. Phys., 16(11), 6883–6900, doi:10.5194/acp-16-6883-2016, 2016.

Eayrs, C., Li, X., Raphael, M. N. and Holland, D. M.: Rapid decline in Antarctic sea ice in recent years hints at future change, Nat. Geosci., 14(7), 460–464, doi:10.1038/s41561-021-00768-3, 2021.

Ekaykin, A. A., Lipenkov, V. Y., Barkov, N. I., Petit, J. R. and Masson-Delmotte, V.: Spatial and temporal variability in isotope composition of recent snow in the vicinity of Vostok station, Antarctica: implications for ice-core record interpretation, Ann. Glaciol., 35(1), 181–186, doi:10.3189/172756402781816726, 2002.

Fujita, K. and Abe, O.: Stable isotopes in daily precipitation at Dome Fuji, East Antarctica, Geophys. Res. Lett., 33(18), doi:10.1029/2006GL026936, 2006.

Guillevic, M., Bazin, L., Landais, A., Kindler, P., Orsi, A., Masson-Delmotte, V., Blunier, T., Buchardt, S. L., Capron, E. and Leuenberger, M.: Spatial gradients of temperature, accumulation and $\delta$ 18 O-ice in Greenland over a series of Dansgaard–Oeschger events, Clim. Past, 9(3), 1029–1051, 2013.

Hersbach, H., Bell, B., Berrisford, P., Hirahara, S., Horányi, A., Muñoz-Sabater, J., Nicolas, J., Peubey, C., Radu, R., Schepers, D., Simmons, A., Soci, C., Abdalla, S., Abellan, X., Balsamo, G., Bechtold, P., Biavati, G., Bidlot, J., Bonavita, M., De Chiara, G., Dahlgren, P., Dee, D., Diamantakis, M., Dragani, R., Flemming, J., Forbes, R., Fuentes, M., Geer, A., Haimberger, L., Healy, S., Hogan, R. J., Hólm, E., Janisková, M., Keeley, S., Laloyaux, P., Lopez, P., Lupu, C., Radnoti, G., de Rosnay, P., Rozum, I., Vamborg, F., Villaume, S. and Thépaut, J.-N.: The ERA5 global reanalysis, Q. J. R. Meteorol. Soc., 146(730), 1999–2049, doi:https://doi.org/10.1002/qj.3803, 2020.

Jones, M. E., Bromwich, D. H., Nicolas, J. P., Carrasco, J., Plavcová, E., Zou, X. and Wang, S.-H.: Sixty Years of Widespread Warming in the Southern Middle and High Latitudes (1957–2016), J. Clim., 32(20), 6875–6898, doi:10.1175/JCLI-D-18-0565.1, 2019.

Jones, T. R., Roberts, W. H. G., Steig, E. J., Cuffey, K. M., Markle, B. R. and White, J. W. C.: Southern Hemisphere climate variability forced by Northern Hemisphere ice-sheet topography, Nature, 554, 351 [online] Available from: https://doi.org/10.1038/nature24669, 2018.

Jouzel, J.: Calibrating the isotopic paleothermometer, Science (80-. )., 286(5441), 910–911, 1999.

Kino, K., Okazaki, A., Cauquoin, A. and Yoshimura, K.: Contribution of the Southern Annular Mode to variations in water isotopes of daily precipitation at Dome Fuji, East Antarctica, J. Geophys. Res. Atmos., 126(23), e2021JD035397, 2021.

Laepple, T. and Huybers, P.: Ocean surface temperature variability: Large model–data differences at decadal and longer periods, Proc. Natl. Acad. Sci., 111(47), 16682–16687, doi:10.1073/pnas.1412077111, 2014.

Landais, A., Ekaykin, A., Barkan, E., Winkler, R. and Luz, B.: Seasonal variations of 17O-excess and d-excess in snow precipitation at Vostok station, East Antarctica, J. Glaciol., 58(210), 725–733, doi:10.3189/2012JoG11J237, 2012.

Landais, A., Casado, M., Prié, F., Magand, O., Arnaud, L., Ekaykin, A., Petit, J.-R., Picard, G., Fily, M. and Minster, B.: Surface studies of water isotopes in Antarctica for quantitative interpretation of deep ice core data, Comptes Rendus Geosci., 2017.

Lorius, C., Merlivat, L. and Hagemann, R.: Variation in the mean deuterium content of precipitations in Antarctica, J. Geophys. Res., 74(28), 7027–7031, doi:10.1029/JC074i028p07027, 1969.

Masson-Delmotte, V., Hou, S., Ekaykin, A., Jouzel, J., Aristarain, A., Bernardo, R. T., Bromwich, D., Cattani, O., Delmotte, M., Falourd, S., Frezzotti, M., Gallée, H., Genoni, L., Isaksson, E., Landais, A., Helsen, M. M., Hoffmann, G., Lopez, J., Morgan, V., Motoyama, H., Noone, D., Oerter, H., Petit, J. R., Royer, A., Uemura, R., Schmidt, G. A., Schlosser, E., Simões, J. C., Steig, E. J., Stenni, B., Stievenard, M., van den Broeke, M. R., van de Wal, R. S. W., van de Berg, W. J., Vimeux, F. and White, J. W. C.: A Review of Antarctic Surface Snow Isotopic Composition: Observations, Atmospheric Circulation, and Isotopic Modeling*, J. Clim., 21(13), 3359–3387, doi:10.1175/2007JCLI2139.1, 2008.

Schlosser, E., Reijmer, C., Oerter, H. and Graf, W.: The influence of precipitation origin on the δ18O–T relationship at Neumayer station, Ekstrmisen, Antarctica, Ann. Glaciol., 39, 41–48, 2004.

Shao, Z.-G. and Ditlevsen, P. D.: Contrasting scaling properties of interglacial and glacial climates, Nat. Commun., 7, 10951, 2016.

Stenni, B., Scarchilli, C., Masson-Delmotte, V., Schlosser, E., Ciardini, V., Dreossi, G., Grigioni, P., Bonazza, M., Cagnati, A., Frosini, D., Karlicek, D., Risi, C., Udisti, R. and Valt, M.: Three-year monitoring of stable isotopes of precipitation at Concordia Station, East Antarctica, Cryosph., 2016(special issue IPICS 2016), 1–30, doi:10.5194/tc-2016-142, 2016.

Stenni, B., Curran, M. A. J., Abram, N., Orsi, A., Goursaud, S., Masson-Delmotte, V., Neukom, R., Goosse, H., Divine, D. and Van Ommen, T.: Antarctic climate variability on regional and continental scales over the last 2000 years, Clim. Past, 13, 1609–1634, 2017.

Touzeau, A., Landais, A., Stenni, B., Uemura, R., Fukui, K., Fujita, S., Guilbaud, S., Ekaykin, A., Casado, M., Barkan, E., Luz, B., Magand, O., Teste, G., Le Meur, E., Baroni, M., Savarino, J., Bourgeois, I. and Risi, C.: Acquisition of isotopic composition for surface snow in East Antarctica and the links to climatic parameters, Cryosph., 10(2), 837–852, doi:10.5194/tc-10-837-2016, 2016.

Werner, M., Jouzel, J., Masson-Delmotte, V. and Lohmann, G.: Reconciling glacial Antarctic water stable isotopes with ice sheet topography and the isotopic paleothermometer, Nat. Commun., 9(1), 3537, doi:10.1038/s41467-018-05430-y, 2018.

Zhu, F., Emile-Geay, J., McKay, N. P., Hakim, G. J., Khider, D., Ault, T. R., Steig, E. J., Dee, S. and Kirchner, J. W.: Climate models can correctly simulate the continuum of global-average temperature variability, Proc. Natl. Acad. Sci., 116(18), 8728–8733, 2019.

---

## Author Comment (AC1)

**General comment**

Stable water isotopes ($\delta^{18}O$) measure in polar regions like Antarctica are traditionally used to reconstruct past surface air temperature (SAT). However, this relationship is influenced by many parameters like surface elevation, air mass sources or sea surface conditions. Moreover, this relationship varies spatially and over time, and has not been investigates using historical simulations (1850 onward), yet. To tackle this issue, Goursaud Oger et al. investigated the SAT – $\delta^{18}O$ relationship during the historical period in Antarctica using an ensemble of historical climate simulations (1851-2004) performed with the isotope-enabled HadCM3 general circulation model. The found strong SAT and precipitation temporal trends during this period, but only weak trends for $\delta^{18}O$, meaning no significant relationship between SAT and $\delta^{18}O$ over one third of Antarctica. They conclude that the decoupling between $\delta^{18}O$ and SAT occurs primarily because of the impact of autumnal sea ice loss during the simulated warming.

The analyses and idea are simple (in a good way) and well written, making the article easy to read and follow. To better quantify the influence of parameters other than temperature on stable water isotopic composition of ice in Antarctica is an important topic for paleoclimate community, and this article represents an additional valuable contribution to that discussion. This article is worthy for publication in CP after addressing the minor points detailed below, including more in-depth analyses of the circulation of air and moisture masses induced by changes in sea ice.

We thank the reviewer for the time and relevant comments he made. These contributed to improve our manuscript. We also hope that the changes we brought will answer your expectations.

**Major comments**

In terms of analyses, the warm – cold anomalies and the seasonal effects are very interesting and well investigated. On the other hand, in the abstract and the conclusion, the authors talk about the involved variations in moisture transport and air mass intrusions due to sea ice transport. I would expect deeper analyses on this aspect, and not just some general statements in the conclusion section. For example, the winds patterns are shown in Figure 4 but not cited in the text.

We are aware that our study is not exhaustive. More analyses could be made to deepen the explanation of processes behind our results. Especially, a complete investigation on the atmospheric circulation change could be lead, adressing the effect of the different teleconnections, through different modes impacting Antarctica (e.g. as done by Marshall and Thompson (2016) for SAT).

Meanwhile, we looked at the impact of the SAM. Our results were given in Appendix F. We show that the HadCM3 reproduces the impacts of SAM on SAT and P reported in previous studies (Clem et al., 2016; Fogt et al., 2020), i.e. colder and drier conditions in a positive SAM. For $\delta^{18}O$, the HadCM3 simulates a depletion in most areas of the Antarctic continent while the SAM is in a positive phase, but these results are associated with relatively low correlation coefficients with means of -0.26±0.11 over the Historical period and -0.27±0.12 for the period 1950 – 2004. We thus conclude that our simulations cannot establish a robust link between the SAM and the Antarctic precipitation weighted $\delta^{18}O$. This result is supported by the diversity of $\delta^{18}O$ measurements from precipitation and firn/ice cores on different Antarctic locations (e.g. Vega et al., 2016; Kino et al., 2021; Servettaz, 2022; Dreossi et al., 2023). Moreover, it was shown that SAM impacts are different with the ENSO phases (Wilson et al., 2016), and that other modes affect Antarctic climate (e.g. Shields et al., 2022).

Finally, we suggest that the impact of the atmospheric circulation on Antarctic precipitation weighted δ[18]O for the Historical period will be strickly tackled in another future study. We thus completed Section 5 ("Drivers"):

*"The dynamic processes behind the sea ice extent induced δ18O changes are complex and multiple. Although the Southern Annular Mode, leading mode of the atmospheric variability in the Southern Hemisphere, might explain part of these δ18O simulated changes (Appendix F), a more comprehensive study might investigate the impact of the atmospheric circulation changes."*

In the conclusion, l.291, we replaced:
*"We indentify three processes [...]"* by *"We suggest [...]"*, meaning that an extended study is necessary to check the atmospheric processes at the origin of our simulated results.

In section 2.1, the model and the setup of simulations should described a little bit more. Six historical simulations have been performed. How the authors make these simulations a little different from each other? With different initial conditions? By changing the value of a parameter? For the HadCM3 model, it should be stated from this section that this is a atmosphere-ocean coupled model.

We only altered the initial conditions, starting each simulation a year apart. We clarified these features and detailed the applied protocol, as asked by the second reviewer l.79 to 90 :

*"Here, we use the Hadley Center Atmosphere-Ocean general circulation model (HadCM3; AOGCM), to run six transient Historical simulations. HadCM3 is a version of the coupled Atmosphere-Ocean UK Met Office climate model (Pope et al., 2000; Gordon et al., 2000). The model is equipped with stable water isotopes (Tindall et al., 2009). Its horizontal resolution is 3.75° × 2.5°, and there are 19 vertical levels (Pope et al., 2000; Gordon et al., 2000; Tindall et al., 2009). The setup of the Historical simulations is described in (Schurer et al., 2014), and follows the recommendations of the third Paleoclimate Modelling Intercomparison Project (PMIP3; Schmidt et al., 2011)(PMIP3; Schmidt et al. 2012). Each simulation is forced with time-varying orbital, solar, volcanic, land-use and well-mixed greenhouse gas forcing. Changes in orbital parameters were calculated following (Berger, 1978). Volcanic forcing is that described in (Crowley et al., 2008). The solar forcing follows (Shapiro et al., 2011). Changes in $CO_2$, $N_2O$ and $CH_4$ were set following the PMIP3 standard (Schmidt et al., 2011). Changes in the abundances of 6 Halocarbons were prescribed following (Tett et al., 2007). Changes in land-cover were prescribed by reclassifying the Global land cover reconstruction developed by (Pongratz et al., 2008). Each of our simulations were only altered by starting each simulation a year apart."*

Also, parameterization relative to isotopes with sea ice should be described? How sea-ice - atmosphere exchanges are taken into account in the model for the isotopes? How ocean free vs. sea ice covered areas considered for the calculation of isotope concentration in surface water vapor? Is there any sublimation of snow on sea ice? With a fractionation effect? Or is there nothing specific coded for the isotope sides, meaning that isotopes are influenced by mainly by changes in air mass and moisture transport, only.

Sea ice is represented in the model and is calculated by the oceanic component. Ice sheets and sea ice in the model are initialised with a δ18O value of -30 and -2 ‰ respectively. The isotope component of HadCM3 ignores the small fractionation associated with sea ice processes and thus makes the approximation that sea ice melting/formation is non-fractionating (Holloway et al. 2016; Tindall et al., 2009; Pfirman et al., 2004). Because of the slow diffusivity of heavy isotope species within ice, sublimation from sea ice is also assumed to be non-fractionating (Jouzel, 1986).

HadCM3 requires a small water and isotope flux to represent iceberg calving and close the hydrological and isotope budgets. Since the isotope flux was not calculated directly from the model, a very small drift in ocean isotopes remains, similar to the case for salinity (Pardaens et al. 2003). The drift is not large enough to affect the results of the century-scale simulations shown here.

The figures need to be improved. The font size of the titles, letters, equations, and labels are really too small in all the figures. The figure 2 needs big improvement. See for example the figures 2 and 5 in the recent paper from Servettaz et al. (https://doi.org/10.5194/tc-17-5373-2023). Moreover, it would help to note somewhere the name of the different regions of Antarctica, and to which colors they do correspond. In the current state of the paper, it is hard to follow which region is where for people not familiar with Antarctica geography.

We increased the fontsize of written text (label, etc) in the figures. We also relocated the subplots of Figure 2 and added the names of the regions. We hope that these improvements will help the reader to follow which region is where.

Note that while recomputing our regional $\delta^{18}O$ trends, we realised that we had made a mistake not specifying those that were not signficant (p-value>0.05). As a result, over the last 50 years simulated by HadCM3, only three regions, the Indian, the Weddell and the Dronning Maud Land coastal regions display significant linear relionships.
We thus adapted Figure 2 by shading in grey non significant trends and completed the caption:
"Grey shaded rows correspond to non significant relationships (p-value>0.05)."
In the text, we removed l. 185 to 187:
*"Over the last fifty years, a part from the Victoria Land where a very weak trend appear, other regions present weaker trends with correlation coefficients now ranging from 0.11 to 0.38 while gradients increase with values ranging from 0.03 ‰ per decade for the WAIS and the plateau, to 0.14 ‰ per decade for the Weddell coast."*
And instead we have written l.185 to 187:
*"Over the last fifty years, only three regions, the Indian, the Weddell and Dronning Maud Land coastal regions keep on displaying significant δ18O trends, that double or more compared to the Historical period, with gradients of 0.08, 0.08 and 0.14 ‰ per decade respectively."*
Also, we adapted the comparison with the results from Casado et al. (2023),
*"They found gradients with the same range of values, from 0.09 ‰ for the Indian coast, to 0.19 ‰ for the Weddell coast, while they found significant relationships where we do not, for time windows varying from 40 to 65 years. Note that for most of the regions, the significance of our simulated relationships disappear when taking time windows lower than 75 years (Appendix D). This could be explained either by a two low simulated anthropogenic variability, as suggested by (Casado et al., 2023), or a change of the drivers on δ18O."*

**Minor comments**

Lines 5, 30, 58, 68, and others: I would replace "water stable isotopes" by "stable water isotopes"
Done.

3$^{rd}$ paragraph of the introduction: Cite also Servettaz et al., TC, 2023 (https://doi.org/10.5194/tc-17-5373-2023)
We added the citation l.51 *"These other controls include: changes related to atmospheric dynamics, such as changes in the synoptic and seasonal nature of precipitation (van Ommen and Morgan, 1997; Krinner and Werner, 2003; Jouzel et al., 2003; Sime et al., 2008; Servettaz et al., 2023)"*.

Lines 46-47 and 92: "Antarctica2k" without space like in Stenni et al. (2017). Define A2k for Antarctica2k here, too.

Done.

Lines 50-53: please add the study from Buizert et al., Science, 2021 (https://doi.org/10.1126/science.abd2897) et Cauquoin et al. CP, 2023 (https://doi.org/10.5194/cp-19-1275-2023), except if you want to focus on warm period only (in that case, some references like Werner et al. (2018) should be removed). If you choose the latter option, please precise at the beginning of the paragraph that you talk here only about warm periods. :

Done, l.48 to 53:

*"These other controls include: changes related to atmospheric dynamics, such as changes in the synoptic and seasonal nature of precipitation (van Ommen and Morgan, 1997; Krinner and Werner, 2003; Jouzel et al., 2003; Sime et al., 2008; Servettaz et al., 2023) and air mass sources (Landais et al., 2021), various impacts from changes in Antarctic ice sheet morphology (Holloway et al., 2016; Werner et al., 2018; Buizert et al., 2021; Goursaud et al., 2021), and sea ice variability (Holloway et al., 2018; Cauquoin et al., 2023)."*

Line 101: please describe briefly the ECHAM5-wiso simulation (AGCM at T106 resolution nudged to ERA40/ERA-Interim). And cite Werner et al. (2011, https://doi.org/10.1029/2011JD015681).

Following a comment of the second reviewer, we replaced the analyses that were extracted from the ECHAM5-wiso simulation published in Stenni et al. (2017), by new analyses processed using an ECHAM6-wiso simulation communicated by Alexandre Cauquoin.

Compared to ECHAM5-wiso, the performance of the water isotopes in ECHAM6-wiso (Stevens et al., 2013, Cauquoin et al., 2019) is clearly improved. This is attributed to: (i) a modification of the supersaturation parameters ; (ii) that the kinetic fractionation at the evaporation over oceans is now assumed to be independant of the wind speed in order to better represent the d-excess versus deuterium relationship from the Antarctic Snow reported by Masson-Delmotte et al. (2008) ; and finally (iii) that the sublimation processes now accounts for the isotopic content of snow over sea ice. Based on the evaluation of global simulations against ERA-interim and ERA5 reanalyses, Cauquoin and Werner (2021) report that the nudging does not significantly change the simulated isotope values, while increasing the resolution generally improves the performance of the simulations. However, the evaluation of the simulated water stable isotopes in precipitation over Antarctica remains rather qualitative (Figure 1, Cauquoin and Werner, 2021).

Having obtained this new model output data from the newer version of ECHAM, we performed the same analysis as previously applied to ECHAM5 and HadCM3. This does indeed resolve the discrepancy between the models – ECHAM6-wiso and HadCM3 (in the newer ECHAM6 version) have equivalent SAT-$\delta^{18}$O surface air temperature relationships.

We thus made the following changes in the text:

- In section 2 ("the data and methods"), l.102 to 110:

*"Our Historical SAT–$\delta18O$ linear relationship at the regional scale are compared with the regional slopes and correlation coefficients that we computed from the AGCM ECHAM6-wiso equipped with water stable isotopes (Cauquoin et al., 2019). The water stable module of this last generation of the model ECHAM was updated compared to its predecessor, especially (i) the supersaturation parameters, (ii) the kinetic fractionation at the evaporation over oceans, now assumed to be independent of the wind speed in order to better represent the d-excess versus deuterium relationship from the Antarctic Snow reported by (Masson-Delmotte et al., 2008), and finally (iii) the sublimation processes now accounting for the isotopic content of snow over sea ice. Here, we use a simulation run at a T127L95 resolution ( 0.9° x 0.9° horizontal resolution and 95 vertical*

*levels) and nudged towards the ERA5 reanalyses (Hersbach et al., 2020) over the period 1979 – 2022 Cauquoin and Werner (2021)."*

- In section 4 ("Temperature versus  δ18O relationships"), l.202:
*"To enable a consideration of model dependency, we also compare our Historical ensemble against a nudged ECHAM6-wiso simulation (Table 1)."*

- In section 4.2 ("Stability over the Historical period and model dependancy"), l.230 to l.237:
*"Interestingly, the ECHAM6-wiso simulation and the last 50 years of our HadCM3 simulation display similar SAT-δ18O relationships. ECHAM6-wiso simulates slightly stronger relationships with a mean correlation coefficient difference of 0.04, while gradients tend to be slightly higher in HadCM3 with a gradient difference of 0.13 ‰/°C. The only notable differences are for Dronning Maud Land and the Indian coast with stronger relationships and higher gradients simulated by HadCM3 (Table 1).*
*Thus, whilst it is unclear whether the nudging of ECHAM6 towards ERA5 reanalysis, the model resolutions or differences in sea ice behaviours, are the main reason for these discrepancies, it is clear that simulated temperature versus δ18O relationships have low but significant uncertainties. These need to be considered, both regionally and for the most relevant climate state, before being undertaking any inferences of past temperatures using isotopes measured in ice cores."*

In section 6 ("conclusions"),  l.284 to 286:
*"Interestingly, we find similar but slightly weaker SAT-δ18O correlations and slightly higher gradients compared to ERA5 –nudged ECHAM6-wiso simulations at the regional scale."*

We also updated Table 1.

Line 115: which reanalysis data?
We precised : « We analyse our simulations against available observations and reanalysis data (e.g. the Climate Forecast System Reanalysis and the the National Centers for Environmental Prediction reanalyses 2) »

Line 158: Medley and Thomas (2019)
Done.

Lines 170-171: sounds strange (strongest for one place and highest for another place)
The correlation coefficients (strength of a linear relationship) and the slope of the relationship (low or high relationship) have no link. One linear relationship can be high (high slope) but weak (low correlation coefficient) which make it not consistent, or it can be low (low slope) but strong (strong correlation coefficient) so the two variables are strongly linear linked through a slow which is low. I hope it makes it sense.

Line 179: Rephrase the beginning of the sentence to avoid the double use of "values".
Done : « They found gradients with the same range of values »

Line 191: ECHAM5-wiso
Done.

Lines 226 and 227: remove the brackets for the two references.
Done

Lines 228-231: maybe the atmosphere-ocean coupling, including sublimation of snow on sea ice?
You are true that differences between model physics could also explain the differences so we

completed l.242 to 245:

*"Thus, whilst it is unclear whether the nudging of ECHAM6 towards ERA5 reanalysis, the model resolutions, the model physics or differences in sea ice behaviours, are the main reason for these discrepancies, it is clear that simulated temperature versus δ18O relationships have low but significant uncertainties."*

Line 236: add a reference to Figure 5.
Done

Line 244: the largest (no capital T).
Done.

Lines 244-246: say explicitly the months.
For clarity, we added in the above line (243) :
*"displaying seasonal anomalies (for the winter season, e.g. from June to August, and for the summer season, e.g. from December to February)"*

Lines 253: there is no figure 5h.
We corrected to Figures cf.

Line 278: It's not ERA4 but ERA40 and ERA-Interim.
Following the changes related to the ECHAM simulations, this sentence was removed.

Lines 279-282: see first major comment. I think more analyses of wind patterns (for example) to demonstrate these conclusions would improve the paper.
Please refer to our response to your first major comment.

Line 294: See second major comment. These specificities of HadCM3 should be said before in the paper.
It was added as asked in your second major comment.

Line 295: higher (no capital H).
Done.

Figure 1: increase the font size (for all figures), including for the equations, make the average and linear regression curves thicker, use something like a_all, a_recent, r_all, r_recent to differentiate the two equations in each plot.
We increased the fontsizes throughout the manuscript. However, we could not add something like " a_all, a_recent, r_all, r_recent", as there is not enough space on one line and that it would hide part of the plot on multiple lines.

Figure 4: state in the legend that these anomalies relative to annual mean. The wind fields are visible. Put more space between the arrows and draw them thicker and bigger. There is a typo ">" in the last line of the legend.

Done.

References.

Cauquoin, A. and Werner, M.: High-Resolution Nudged Isotope Modeling With ECHAM6-Wiso: Impacts of Updated Model Physics and ERA5 Reanalysis Data, J. Adv. Model. Earth Syst., 13(11), e2021MS002532, doi:https://doi.org/10.1029/2021MS002532, 2021.

Clem, K. R., Fogt, R. L., Turner, J., Lintner, B. R., Marshall, G. J., Miller, J. R. and Renwick, J. A.: Record warming at the South Pole during the past three decades, Nat. Clim. Chang., 10(8), 762–770, 2020.

Holloway, M.D., Sime, L.C., Singarayer, J.S., Tindall, J.C., & Valdes, P.J. (2016). Reconstructing paleosalinity from δ18O: Coupled model simulations of the Last Glacial Maximum, Last Interglacial and Late Holocene. *Quaternary Science Reviews, 131*, 350-364.

Jouzel, J. (1986), Multiphase and multistage condensation processes Hand-book of Environmental Isotope Geochemisty, vol. 2,The Terrestrial Environment, pp. 61–112, Elsevier, New York

Marshall, G. J., & Thompson, D. W. (2016). The signatures of large-scale patterns of atmospheric variability in Antarctic surface temperatures. *Journal of Geophysical Research: Atmospheres, 121*(7), 3276-3289.

Pardaens, A., H. Banks, J. Gregory, and P. Rowntree (2003), Freshwater transports in HadCM3, Clim. Dyn., 21(2), 177–195.

Pfirman, S., W. Haxby, H. Eicken, M. Jeffries, and D. Bauch (2004),Drifting Arctic sea ice archives changes in ocean surface conditions,Geophys. Res. Lett.,31, L19401, doi:10.1029/2004GL020666.

Tindall, J. C., P. J. Valdes, and L. C. Sime (2009), Stable water isotopes in HadCM3: Isotopic signature of El Nin õ–Southern Oscillation and the tropical amount effect,J. Geophys. Res.,114, D04111, doi:10.1029/2008JD010825.

Turner, J., Connolley, W., Cresswell, D., & Harangozo, S. (2001). The simulation of Antarctic sea ice in the Hadley Centre climate model (HadCM3). *Annals of Glaciology, 33*, 585-591.

Vega, C. P., Schlosser, E., Divine, D. V., Kohler, J., Martma, T., Eichler, A., ... & Isaksson, E. (2016). Surface mass balance and water stable isotopes derived from firn cores on three ice rises, Fimbul Ice Shelf, Antarctica. *The Cryosphere, 10*(6), 2763-2777.

---

## Author Comment (AC2)

The manuscript employs simulations generated by an isotope-enabled General Circulation Model (GCM) to assess the temporal variations in isotopic composition, surface temperature, precipitation, and sea ice concentration within Antarctica over the past 200 years. Specifically, they suggest that differences between the simulated isotopic composition and temperature variations can be attributed to change of precipitation patterns due to the impact of sea ice concentration on the moisture pathways toward Antarctica.

The topic addressed by the manuscript is extremely important because the accuracy of the isotopic paleothermometer is directly affected by the link between isotopic composition in ice cores and temperature, which is only constrained empirically. Models are invaluable tools to study and explore the relationship between isotopic composition and temperature spatially, temporally, and with the different time scales. Here, the study based on the outputs of a single model (with occasional comparison with results from other studies using another one) lacks robustness to provide concrete evidence that can help strengthen our understanding of the isotopic paleothermometer. While the study suggested by the authors is worth pursuing, several shortcomings weaken what could otherwise be an important study for the field.

Many thanks for your time spent on providing this careful, thorough and very valuable review. We think it has very substantially improved our manuscript.

**General comments:**

The manuscript predominantly relies on the outcomes of multiple runs from the HadCM3 model. Although the authors note in the Method section that 'HadCM3 provides a reasonable representation of Antarctic Climate and δ18O,' there is a notable absence of a critical evaluation of the model's performance in comparison to available observations in Antarctica. The manuscript does not provide values indicating potential biases or errors in the model. It is challenging to assess the trustworthiness of the model outputs without a direct comparison with present-day observational data in HadCM3 iso, similar to the approach in Figure 1 of (Werner et al., 2018). This becomes more significant as the manuscript critiques observations without addressing potential biases in the studied model, even though the IPCC specifically relies on a diverse ensemble of models to balance potential biases. While a few articles are referenced (specifically (Holloway et al., 2016; Tindall et al., 2009)) that offer some comparisons with discrete values, the absence of confidence intervals makes it difficult to evaluate the robustness of the relationship between isotopic composition and temperature. This limitation affects the confidence in the authors' results, particularly given the extensive literature containing data that could be directly compared with the model outputs for temperature data (Jones et al., 2018)(Jones et al., 2018), isotopic composition (Dittmann et al., 2016; Ekaykin et al., 2002; Landais et al., 2017; Masson-Delmotte et al., 2008; Schlosser et al., 2004; Stenni et al., 2016; Touzeau et al., 2016), NSIDC products for the sea ice…

This is a good point. In response, we have added a new careful evaluation of the model. The new appendix (Appendix A), provide an evaluation of the simulated Antarctic Surface Air Temperature (SAT), precipitation (P) and precipitation weighted $\delta^{18}O$. The results show as expected a warm bias in the Antarctic interior – this is also observed in other models such as in Polar WRF (Zhang et al., 2022); and a dry bias in coastal regions. Overall HadCM3 performs roughly in line with expectations derived from other similar models, and have a reasonable representation of Antarctic surface climate and $\delta^{18}O$.

In the text, we referred to the appendix l.94 to l.96: *"HadCM3 provides a reasonable representation of Antarctic climate and δ18O (Appendix A, as well as Turner et al., 2006; Tindall et al., 2009; Holloway et al., 2016)."*
Appendix A can be found from page 26:

*Appendix A: HadCM3 evaluation of Antarctic surface climate and δ₁₈O*

*A1 Method*

[revised manuscript text omitted]

Some of the results are compared with outputs from another General Circulation Model (GCM), specifically ECHAM5wiso, to assess 'model dependency.' This comparison reveals substantial differences, with predicted slopes between isotopic composition and temperature regionally or across all of Antarctica showing almost a 100% disparity. These findings contribute to a diminishing confidence in the outputs of the HadCM3 model. It's worth noting that the comparison involves an older version of ECHAM with isotopes (ECHAM6wiso, released in 2021, has been demonstrated to be more accurate according to (Cauquoin and Werner, 2021)). Additionally, ECHAM5wiso was nudged with ERA40, a choice that may seem unconventional given the availability of more recent products like ERA-interim in 2006 (Dee et al., 2011) and ERA5 in 2018 (Hersbach et al., 2020).

Following the suggestions above, we replaced the analysis with the latest generation of the AGCM ECHAM equipped with water isotopes: ECHAM6-wiso (Stevens et al., 2013, Cauquoin et al., 2019). As stated by the reviewer, compared to ECHAM5-wiso, the performance of the water isotopes in ECHAM6-wiso is clearly improved. This is attributed to: (i) a modification of the supersaturation parameters ; (ii) that the kinetic fractionation at the evaporation over oceans is now assumed to be independent of the wind speed in order to better represent the d-excess versus deuterium relationship from the Antarctic Snow reported by Masson-Delmotte et al. (2008) ; and finally (iii) that the sublimation processes now accounts for the isotopic content of snow over sea ice. Based on the evaluation of global simulations against ERA-interim and ERA5 reanalyses, Cauquoin and Werner (2021) report that the nudging does not significantly change the simulated isotope values, while increasing the resolution generally improves the performance of the simulations. However, the evaluation of the simulated water stable isotopes in precipitation over Antarctica remains rather qualitative (Figure 1, Cauquoin and Werner, 2021).

Having obtained this new model output data from the newer version of ECHAM, we performed the same analysis as previously applied to ECHAM5 and HadCM3. As implied by the reviewer, using the newer version of the ECHAM indeed entirely resolve the discrepancy between the models – ECHAM6-wiso and HadCM3 (in the newer ECHAM6 version) now have equivalent SAT-$\delta^{18}$O surface air temperature relationships.

We thus made the following changes in the text:
- In section 2 ("Data and methods"), l.106 to 114:
*"Our Historical SAT–δ18O linear relationship at the regional scale are compared with the regional slopes and correlation coefficients that we computed from the AGCM ECHAM6-wiso equipped with water stable isotopes (Cauquoin et al., 2019). The water stable module of this last generation of the model ECHAM was updated compared to its predecessor, especially (i) the supersaturation parameters, (ii) the kinetic fractionation at the evaporation over oceans, now assumed to be independent of the wind speed in order to better represent the d-excess versus deuterium relationship from the Antarctic Snow reported by (Masson-Delmotte et al., 2008), and finally (iii) the sublimation processes now accounting for the isotopic content of snow over sea ice. Here, we use a simulation run at a T127L95 resolution ( 0.9° x 0.9° horizontal resolution and 95 vertical levels) and nudged towards the ERA5 reanalyses (Hersbach et al., 2020) over the period 1979 – 2022 Cauquoin and Werner (2021)."*

- In section 4 ("Temperature versus  δ18O relationships"), l.211:
*"To enable a consideration of model dependency, we also compare our Historical ensemble against a nudged ECHAM6-wiso simulation (Table 1)."*

- In section 4.2 ("Stability over the Historical period and model dependancy"), l.242 to l.253:

*"Interestingly, the ECHAM6-wiso simulation and the last 50 years of our HadCM3 simulation display similar SAT-δ18O relationships. ECHAM6-wiso simulates slightly stronger relationships with a mean correlation coefficient difference of 0.04, while gradients tend to be slightly higher in HadCM3 with a gradient difference of 0.13 ‰/°C. The only notable differences are for Dronning Maud Land and the Indian coast with stronger relationships and higher gradients simulated by HadCM3 (Table 1). Thus, whilst it is unclear whether the nudging of ECHAM6 towards ERA5 reanalysis, the model resolutions or differences in sea ice behaviours, are the main reason for these discrepancies, it is clear that simulated temperature versus δ18O relationships have low but significant uncertainties. These need to be considered, both regionally and for the most relevant climate state, before being undertaking any inferences of past temperatures using isotopes measured in ice cores."*

In section 6 ("conclusions"),  l.307 to 308:
*"Interestingly, we find similar but slightly weaker SAT-δ18O correlations and slightly higher gradients compared to ERA5 –nudged ECHAM6-wiso simulations at the regional scale."*

Table 1 is updated to reflect the replacement of ECHAM5 with ECHAM6 output.

**Table 1. Historical SAT–$\delta^{18}$O relationships at the regional scale.** Slope (in ‰/°C) plus or minus the standard error, and the correlation coefficient (into brackets) of the surface-weighed average of surface air temperature against the surface-weighed average of $\delta^{18}$O for the Antarctic regions as defined in the PAGES Antarctica2k project (Stenni et al., 2017): the plateau, the Indian coast, the Weddell coast, the Peninsula, the WAIS, Victoria Land and Dronning Maud Land, simulated by the ECHAM6-wiso model (, over the period 1979-2022, 44 points, 'ECHAM6-wiso') and simulated by HadCM3 over he last 50 years (1955-2004, 50 points, 'last 50 years of HadCM3'), and over the whole historical simulated period (1851-2004, 154 points, 'Historical HadCM3') using the ensemble mean of the six simulations (see methods). All the relationships are significant (p-values<0.05).

|  | ECHAM6-wiso | last 50 years of HadCM3 | Historical HadCM3 |
|---|---|---|---|
| Plateau | 0.48±0.07 [0.71] | 0.61±0.14 [0.52] | 0.57±0.07 [0.53] |
| Indian coast | 0.29±0.08 [0.48] | 0.55±0.15 [0.46] | 0.67±0.07 [0.59] |
| Weddell coast | 0.49±0.11 [0.57] | 0.57±0.11 [0.59] | 0.57±0.07 [0.57] |
| Peninsula | 0.37±0.05 [0.74] | 0.28±0.06 [0.52] | 0.31±0.02 [0.71] |
| WAIS | 0.56±0.07 [0.75] | 0.60±0.12 [0.58] | 0.50±0.05 [0.61] |
| Victoria Land | 0.43±0.13 [0.46] | - | 0.30±0.12 [0.19] |
| Dronning Maud Land | 0.43±0.13 [0.46] | 0.76±0.12 [0.69] | 0.49±0.05 [0.60] |
| West Antarctica | 0.49±0.11 [0.59] | 0.50±0.10 [0.57] | 0.70±0.07 [0.62] |
| East Antarctica | 0.48±0.08 [0.69] | 0.49±0.10 [0.57] | 0.56±0.06 [0.58] |
| All Antarctica | 0.45±0.09 [0.59] | 0.67±0.13 [0.60] | 0.57±0.06 [0.62] |

Overall, the manuscript does not engage in a critical discussion regarding the capabilities of GCMs (CMIP5 or CMIP6) to accurately depict regional-scale climate dynamics. This is noteworthy given the existing body of literature highlighting that the simulated variability often underestimates (Laepple and Huybers, 2014; Shao and Ditlevsen, 2016; Zhu et al., 2019). It would be beneficial for the manuscript to address the impact of discrepancies between modelling outputs and observations within this context. A thoughtful consideration of this aspect would enhance the argument and bolster confidence in the assertion that the results from HadCM3 are reliable in this context.

This is another valuable point. For reasons of focus and space we cannot engage in a particularly large discussion regarding the capabilities of GCMs (CMIP5 or CMIP6) to accurately depict regional-scale climate dynamics. However, as implied in this comment, the manuscript is improved by some engagement with the body of literature highlighting that the simulated variability often underestimated (Laepple and Huybers, 2014; Shao and Ditlevsen, 2016; Zhu et al., 2019).

It would be beneficial for the manuscript to address the impact of discrepancies between modelling outputs and observations within this context. A thoughtful consideration of this aspect would enhance the argument and bolster confidence in the assertion that the results from HadCM3 are reliable in this context.

We added add suggestions for further comparisons with observations towards the end of the Section 6 ("Conclusion"), l.329 to 332: *"Finally, more stable water isotope records from Antarctic ice and firn core data are more than needed to evaluate models, as well as to lead model-data investigations of past climates, comparing SAT–δ18O relationships from different water stable isotopes enabled model, in line with the work of the Stable Water Isotope Intercomparaison Group 2 (SWING) (Risi et al., 2012)."*

The comparison with observational products seems to selectively draw from publications that align with the authors' findings, potentially overlooking contributions that present contrasting results (Clem et al., 2020; Jones et al., 2019) or dismissing them without explicit justification (Casado et al., 2023), in which I was involved as a lead author). There are several instances detailed below in the specific comments when the authors obtained different results using the same calculation (trends from linear regression) and presumably the same datasets (PAGES 2k) and obtain different results without providing an explanation. Notably, in (Casado et al., 2023), considerable effort was invested in validating trend calculations using the products from (Stenni et al., 2017). It would be valuable for the manuscript to articulate why different trends are identified here, especially as the current narrative suggests a potential error in (Casado et al., 2023). While the scientific method encourages revisiting and improving upon previous results, the manuscript should provide evidence-based arguments when challenging peer-reviewed scientific findings, moving beyond statements like 'It is unclear why the trend in (Casado et al., 2023) is higher."

We welcome the statistical efforts made in your recent study (Casado et al., 2023). Our wordings were inappropriate: we obviously did not mean that your results were wrong but that we were not able to attribute the mismatch. Instead, we should have developed the idea behind the sentence « It is unclear why the trend in Casado et al. (2023) is higher », the revised version of these sentences read, p.6 l.182 to l.185:

*« Casado et al. (2023) provide a higher trend from 1950–2005 of 0.11±0.02 ‰ per decade, based on ice core data. Different reasons could explain that mismatch that we are not able to elucidate so far, inter alia: (i) a model discrepancy to resolve processes, (ii) the model resolution, (iii) the geographical distribution of the ice core locations, (iv) the different methods for the SAT – δ18O calibration. »*

In Section 5 (Drivers of δ18O changes), there appears to be a lack of evidence supporting the hypothesis regarding the mechanisms behind δ 18 O variations. It is commonly acknowledged that correlation does not necessarily imply causation. In this context, the connections between δ18O and other climatic parameters are not based on correlation but on the representation of end-members (warmest and coldest conditions) on a map. The linkage between δ18O and temperature, specifically attributing the decoupling to sea ice concentration as depicted in Figures 4 and 5, is unclear. Notably, the authors seem to have overlooked the potential impact of the SAM, which has been suggested to influence both isotopic composition (Kino et al., 2021) and sea ice concentration around Antarctica (Eayrs et al., 2021).

We are aware that our study is not exhaustive. More analyses could be made to deepen the explanation of processes behind our results. Especially, a complete investigation on the atmospheric circulation change could be lead, addressing the effect of the different teleconnections, through different modes impacting Antarctica, for example, as provided by Marshall and Thompson (2016) for SAT.

To help address this comment, new analysis of the impact of the SAM is given in Appendix G. This shows that HadCM3 reproduces the impacts of SAM on SAT and P reported in previous studies (Clem et al., 2016; Fogt et al., 2020), *i.e.* colder and drier conditions in a positive SAM. For $\delta^{18}$O, HadCM3 simulates depletion in most areas of the Antarctic continent while the SAM is in a positive phase, but these results are associated with relatively low correlation coefficients with means of -0.26±0.11 over the Historical period and -0.27±0.12 for the period 1950 – 2004. We thus conclude that our simulations cannot establish a robust link between the SAM and the Antarctic precipitation weighted $\delta^{18}$O. This result is supported by the diversity of $\delta^{18}$O measurements from precipitation and firn/ice cores on different Antarctic locations (*e.g.* Vega et al., 2016; Kino et al., 2021; Servettaz, 2022; Dreossi et al., 2023). Moreover, it was shown that SAM impacts are different with the ENSO phases (Wilson et al., 2016), and that other modes affect Antarctic climate (*e.g.* Shields et al., 2022). Further analysis on the impact of the atmospheric circulation on Antarctic precipitation weighted $\delta^{18}$O for the Historical period would need to be the subject of a future study. The new results are references in Section 5 ("Drivers") p10 l.292 to l.295 as:

*"The dynamic processes behind the sea ice extent induced δ18O changes are complex and multiple. Although the Southern Annular Mode, leading mode of the atmospheric variability in the Southern Hemisphere, might explain part of these δ18O simulated changes (Appendix G), a more comprehensive study might investigate the impact of the atmospheric circulation changes."*

In the conclusion, l.310, we replaced:
*"We identify three processes [...]"* by *"We suggest [...]"*, meaning that an extended study is necessary to check the atmospheric processes at the origin of our simulated results.

Here is our new Appendix G:

[revised manuscript text omitted]

**Specific comments:**

Lines 5 to 6: "Our ensemble captures observed historical SAT and precipitation trends, and weak δ18O trends." Currently, this statement is not supported within this manuscript. It does conflict with recent publications on the topic (Casado et al., 2023; Clem et al., 2020; Jones et al., 2019) without clear justifications for considering this manuscript's results as more valid than those published in peer-reviewed journals.

*Addressed, please see new Appendix A on the model evaluation over Antarctica.*

Lines 6 to 7: "The weak δ18O trends mean there is no significant relationship between SAT and δ18O over one third of Antarctica, and also half of our considered ice core sites, though relationships are stronger when using regional averages." While it can be debated, it's important to note that in Antarctica, every site where precipitation isotopic composition has been sampled demonstrates a remarkably robust correlation with temperature (Dittmann et al., 2016; Fujita and Abe, 2006; Landais et al., 2012; Schlosser et al., 2004; Stenni et al., 2016; Touzeau et al., 2016). Any counterargument should be accompanied by evidence explaining why the findings in this manuscript should be considered more valid than those published in peer-reviewed journals.

We agree that the published relationships based on precipitation samples all showed a significant correlation between SAT and $\delta^{18}O$. However, they remain few, so we cannot exclude that the absence of SAT – $\delta^{18}O$ derived from firn/ice core (especially for coastal area over the last decades, please see Goursaud et al., 2019) were preserved from the atmosphere. As a few examples no relationship was found in firn/ice cores from Dronning Maud Land, near the Neumayer station

(Vega et al., 2016), in the Ross Sea sector (Bertler et al., 2011), and in Adélie Land, close to Dumont d'Urville (Goursaud et al., 2017).

Lines 19 to 21: "The collapse of Antarctic ice shelves have similarly increased in frequency (Graham et al., 2022; Milillo et al., 2022; Wille et al., 2022), with a 1,600 square kilometers iceberg breaking away from the Brunt ice shelf on January 22nd, 2023." This seems anecdotal and unrelated to the topic of the manuscript.

Addressed, we removed that part of the sentence : *« The collapse of Antarctic ice shelves have similarly increased in frequency (Graham et al., 2022; Milillo et al., 2022; Wille et al., 2022). »*

Lines 26 to 27: "The remote nature of this vast continent means that observational data covering Antarctica are sparse in both space and time (Turner et al., 2004)."
Rewrite, too vague and imprecise. Satellite data are observational, and they are regularly gridded and relatively dense. Also, most of the observations are in the last 50 years, so they are not "sparse in time", but rather extremely concentrated on a specific period. I understand what you meant, but the sentence could be improved.

We updated the manuscript:
*"The relatively short satellite record (since 1979 only), and sparsity of in-situ observational data from Antarctica, mean that reconstruction of past temperature change is important for understanding natural variability, and hence our ability to detect anthropogenic climate change in Antarctica (Turner et al., 2004; Casado et al. 2023)."*

Lines 27 to 28: "The reconstruction of past temperature change is thus of paramount importance for understanding natural variability, versus effects in response to anthropogenic climate change""paramount importance" does not seem appropriate here. "versus effects in response..." is nonsensical.

*Removed, as above.*

Lines 34 to 35: "it is sometimes referred as the 'ice core paleothermometer' (Lorius and Merlivat, 1977; Masson et al., 2000)."
It is referred as the "Isotopic paleothermometer" in the specific references that are cited here. Also Lorius and Merlivat 1977 does not qualify as a peer-reviewed publication, consider replacing by (Lorius et al., 1969). Other proxies from ice cores can be used for past temperature reconstructions, for instance the borehole thermometry.

Addressed « ice core paleothermometer » replaced by « isotopic paleothermometer ». And Lorius et al., 1969 cited instead of Lorius and Merlivat, 1977.

Line 36: "The paleothermometer" is not sufficient, should be describe as "isotopic paleothermometer".

Done.

Lines 38 to 41: "This ice core based record was then used to show that simulated temperatures from the Atmospheric General Circulation (AGCM) models run in the frame of the Coupled Model

Intercomparison Project Phases 5 (Taylor et al., 2012) and 6 (Eyring et al., 2016, CMIP6) are too low. However, the paleothermometer relationship has been shown to vary spatially over the Antarctic continent (e.g. Sime et al., 2008, 2009a)."
(Casado et al., 2023) does not state that the temperatures are not too low, but that the variability (natural and forced) is too low. In addition, variable isotope-temperature conversions are used for the different regions of Antarctica in Casado et al, 2023, so it seems that the use of "however" here suggests an opposition that is not based on actual opposite point of views.

L.37 to l.40, we corrected: *« This ice core based record was then used to show that the simulated temperature variability from the Atmospheric General Circulation (AGCM) models run in the frame of the Coupled Model Intercomparison Project Phases 5 (Taylor et al., 2012) and 6 (Eyring et al., 2016, CMIP6) is too low. »*
Alslo, we removed: « However » : *« The isotopic paleothermometer relationship has been shown to vary spatially over the Antarctic continent (e.g. Sime et al., 2008, 2009a). »*

Line 49: "The geographical variability in the 'paleothermometer' is due to controls on d18O other than SAT." This statement appears slightly misleading in the sense that it doesn't reflect that Rayleigh distillation (Dansgaard et al, 1964) suggests already that the signal acquired across the distillation pathway, and not a pure local surface temperature signal.

Addressed. Sentence now reads, l.48: *« The geographical variability in the 'isotopic paleothermometer' is due to controls on $\delta18O$ other than related to SAT. »* This includes changes of temperatures along the air mass pathways as SAT and condensation temperatures are linked, thus affecting the Rayleigh distillation.

Lines 59 to 62: "AGCM isotopic studies have focused on the effects of external forcing on the SAT–$\delta18O$ relationship, including elevation and greenhouse gases across a range of timescales (e.g. Sime et al., 2009b; Werner et al., 2018; Goursaud et al., 2021). A major result is that, for differing time-scales and driving mechanisms, different SAT–$\delta18O$ relationships can be obtained." This sentence suggests that this result was only found using isotope enabled GCM studies, while several proxy based studies have also shown this, including relatively old ones, see for instance (Guillevic et al., 2013; Jouzel, 1999).

Here, we focused the paragraph on the use of AGCM, and did not mean that such studies were not made using observations. We thus completed the sentence by beginning with « For instance » to make the link with the preceding sentence, l.59 to 61 :
*« For instance, AGCM isotopic studies have focused on the effects of external forcing on the SAT–$\delta18O$ relationship, including elevation and greenhouse gases across a range of timescales (e.g. Sime et al., 2009b; Werner et al., 2018; Goursaud et al., 2021). »*

Lines 78 to 83: "The Historical simulation protocol was defined in the frame of the CMIP Phase 6 (Eyring et al., 2016), with an express purpose to investigate the anthropogenic forcing on climate (Johns et al., 2003) and serve as a benchmark to evaluate model performance (Andrews et al., 2020; Miller et al., 2021; Parsons et al., 2020; Rong et al., 2021; Roach et al., 2020). There have been few examples of studies using Historical simulations focused over Antarctica (Gao et al., 2021; Purich and England, 2021; Raphael et al., 2020; Roach et al., 2020).
Here, we use the Hadley Center general circulation model (HadCM3; GCM), to run six transient Historical simulations."
This is technically correct, it also implies that HadCM3 is a CMIP6 model, although it is actually a CMIP5 model. The description should first be the model. Then be the transient historical simulations that were done. At this point, for non-modeller specialist such as myself, it is not possible to know what the historical simulation protocol entails, so rather than giving examples of

studies, the key important points that a reader should know should be described here. For instance, it is not clear if/how the surface conditions were prescribed since only the atmospheric component was used. Was the sea ice concentration simulated or prescribed? If it was prescribed, since HadGEM performs relatively poorly for sea ice concentration, surface ocean temperature, and to some extent for the 850hPa temperature (Agosta et al., 2015), how is that affecting the results?

The paragraph is re-ordered as requested. Parts relating to CMIP6 are removed, to prevent confusion, and instead the details of the protocol are given, as requested, after the model description. This includes, as requested some more information on the HadCM3 coupled Atmosphere-Ocean model, and that sea-ice is not prescribed but calculated. The paragraph dedicated to our model description and simulations is now p.3 l.79 to 90:
*"Here, we use the Hadley Center Atmosphere-Ocean general circulation model (HadCM3; AOGCM), to run six transient Historical simulations. HadCM3 is a version of the coupled Atmosphere-Ocean UK Met Office climate model (Pope et al., 2000; Gordon et al., 2000), which means that sea ice is prognostic. The model is equipped with stable water isotopes (Tindall et al., 2009). Its horizontal resolution is 3.75° × 2.5°, and there are 19 vertical levels (Pope et al., 2000; Gordon et al., 2000; Tindall et al., 2009). The setup of the Historical simulations is described in (Schurer et al., 2014), and follows the recommendations of the third Paleoclimate Modelling Intercomparison Project (PMIP3; Schmidt et al., 2011)(PMIP3; Schmidt et al. 2012). Each simulation is forced with time-varying orbital, solar, volcanic, land-use and well-mixed greenhouse gas forcing. As above, sea ice is not prescribed, rather calculated by the model. Changes in orbital parameters were calculated following (Berger, 1978). Volcanic forcing is that described in (Crowley et al., 2008). The solar forcing follows (Shapiro et al., 2011). Changes in CO2, N2O and CH4 were set following the PMIP3 standard (Schmidt et al., 2011). Changes in the abundances of 6 Halocarbons were prescribed following (Tett et al., 2007). Changes in land-cover were prescribed by reclassifying the Global land cover reconstruction developed by (Pongratz et al., 2008). Each of our simulations were only altered by starting each simulation a year apart."*
Lines 89 to 90: "HadCM3 provides a reasonable representation of Antarctic climate and δ18O (Turner et al., 2006; Tindall et al., 2009; Holloway et al., 2016)." Tindall et al, 2009 provides a comparison of d18O with observations from mostly tropical and temperate regions, with only 2 data points in Antarctica. Holloway et al, 2016 provides a comparison of the outputs of the model and 4 ice core records during the last glacial maximum. To my knowledge, no comparison between modern δ 18 O from observations and model outputs has been published. Considering the large warm bias that most of the isotope enabled CMIP5 models suffered which have been fixed in CMIP6 versions (Cauquoin and Werner, 2021; Werner et al., 2018), it feels like this statement does not provide the necessary information to know if we can trust the outputs during the historical periods. Please consider reproducing Figure 1 of (Werner et al., 2018).

Please refer to our above response related to the model evaluation over Antarctica. This is now added in a new Appendix A.

Lines 96 to 97: "Where we regress climate variables against δ18O, the linear regressions are computed using the stacked individual ensemble members, rather than using the ensemble mean." Unclear to me. Are you computing the linear regression on a stack of all the individual members ? Or are you computing it against individual members and then stacking the linear regression ? The former seems fairly similar than using the ensemble mean.

We stacked all the members for each climate variable, and then processed linear regressions (thus at the annual scale, on 153 x 6 = 918 points). This is now clarified l.102-104 : « *Where we regress climate variables against δ18O, the linear regressions are computed using the stacked individual ensemble members, rather than using the ensemble mean. This approach ensures that the ensemble*

*variability is included in our linear regression statistics and increased the number of points on which the regressions are processed. »*

Lines 99 to 101: "Our Historical SAT–d18O linear relationship at the regional scale, as well as at the nearest model grid-cell to each ice core location, are compared with the ECHAM5-wiso slopes and correlation coefficients provided in Stenni et al. (2017)." Why would you use ECHAM5-wiso here when your manuscript is about HadCM3 ? If you're using an isotope enabled version of ECHAM, why not use ECHAM6 which has been released in 2021. And if the goal is to provide a reference that is published, why is the comparison with observations not included, i.e. (Casado et al., 2017; Fujita and Abe, 2006; Masson-Delmotte et al., 2008; Schlosser et al., 2004; Stenni et al., 2016; Touzeau et al., 2016).

Addressed. Following the reviewers excellent suggestion, we have switched from using the older ECHAM5 results to those from ECHAM6. This does indeed solve many problems in the previous version of the manuscript.  As useful implied above, we also add suggestions for further comparisons with observations towards the end of the Section 6 ("Conclusion"), l.329 to 332: *"Finally, more stable water isotope records from Antarctic ice and firn core data are more than needed to evaluate models, as well as to lead model-data investigations of past climates, comparing SAT–δ18O relationships from different water stable isotopes enabled model, in line with the work of the Stable Water Isotope Intercomparaison Group 2 (SWING) (Risi et al., 2012)."*

Line 114: "3 Trends in Antarctic SAT, precipitation, sea ice and δ18O" Is this a result section ? it seems to include results and discussion, but then section 4 and 5 as well. While I don't think Climate of the Past has a strict rule on which structure to use for manuscript, I am not convinced that the classical structure wouldn't help the readability of the manuscript.

Thank you for pointing out that there is a lack of signposting of the Results.  To address this, we create a new single Results section, out of the previous three results subsections. Because these previous results sections now become subsections of the single Results section, this also necessitates removing subsubsubsection heading in what is now 3.1.2, because they would be at the three decimal place subsubsubsection level, which is not permitted. We agree with the reviewer that overall this change improves the structure and readability of the manuscript. To help with signposting we also add a short overview of the Results section: *"This section uses these model data and methods to examine: trends in Antarctic SAT, precipitation, sea ice and δ18O, including at the continental and regional scale; relationships between temperature versus δ18O, including their stability, and model dependency; and finally, the drivers of δ18O changes."*

Lines 122 to 123: "This is consistent with observations of 0.12±0.07 °C per decade over 1957-2006 (Steig et al., 2009) and 0.11±0.08 °C per decade over 1959-2012 (Nicolas and Bromwich, 2014)." This is only partially true. Jones et al, 2019 reports larger warming across Antarctica when the SAM-congruent trend has been taken into account. Clem et al, 2019 reports temperature increase of 0.6°C per decades at the south pole station, where the map in Figure 1E. reports actually a cooling between 1850-1900 and 1950-2000. ERA5 reports as well a large warming across Antarctica, which is indeed a reanalysis based on satellite observations.
Also, is the comparison really accurate if you compare on the one hand model outputs over all of Antarctica and meteorological observations from N&B which are clearly biased toward coastal regions?

Steig et al. (2009) and Nicholas and Bromwich (2014) were cited to aid comparison with Antarctic-wide SAT trends. Adding the Jones et al. (2019) reference is a very useful suggestion, given they also focus on West Antarctica and Antarctica Peninsula. Given they show that the (positive) trends are the highest for the station located on Antarctic Peninsula, this is added in Section 3.2.1, l.163:

*"At the scale of station locations, (Jones et al., 2019) also show the highest trends for the Peninsula."*

Thank you to the referee for also pointing the study of Clem et al. (2019). Clem et al. (2019) report an increase of 0.6°C per decade at the south pole station over the period 1989–2018, with record-high annual SAT in 2002, 2009, 2013 and 2013, reflecting a very recent trend (also made clear on Fig. 1c). They attribute this warming to an increase in northerly winds at the South Pole (Fig. 2). For the period 1957–2002 (Fig. 1c), they observe negative trends. They also display that SAT trends simulated by CMIP5 models at South Pole, are lower for the pre-industrial period compared to the historical period (Fig3a). Although these results are valuable, in this case we do not include it in the study as we did not focus on single sites, rather at the regional to continental scale.

Lines 129 to 130: "Forecast System Reanalysis (CFSR), and 7.1±1.5 mm/y per decade from the National Centers for Environmental Prediction reanalyses 2 (NCEP-2) over 1979-2009."
It seems that different sources of data (observations, reanalyses, or other type of models) are used for different variables. Wouldn't it be valuable to compare all of your variables with one systematic source of data, may it be direct observations, satellite observations, reanalyses…

Agreed, however unfortunately, we are not aware of a single source of measurements or reanalysis product that encompass all addressed variables, particularly the water isotopes. Also, given different classes of data have different types of uncertainties associated, there is also value in comparing to more than one data type.

Lines 166 to 168: "Despite the simulated increases in SAT and precipitation, d18O shows a very weak trend of 0.04 ±0.003 ‰ per decade (r=0.21) over the last 50 years. Interestingly, (Casado et al., 2023) provide a higher trend from 1950–2005 of 0.11± 0.02 ‰ per decade. It is unclear why the trend in (Casado et al., 2023) is higher." The response provided seems insufficient, particularly in light of the extensive sensitivity tests conducted in (Casado et al., 2023)to elucidate the disparities with the trends observed in S(Stenni et al., 2017). It would be helpful to have clarification on the handling of isotopic data from the PAGES2k network in this context, such as the methodology for averaging monthly isotopic data with annual and interannual data. Given that a trend is essentially a mathematical representation, and both this manuscript and Casado et al., 2023 utilise the same dataset, it raises concerns about the disparity in values. Moreover, the methods employed in Casado et al., 2023 were replicated using the outputs generated in Stenni et al., 2017, resulting in a slope of 0.10 permil per decade. Additionally, an alternative method based on dynamical system theory yielded an even larger value.

The $\delta^{18}O$ trends given here are those simulated by HadCM3. We attempted to clarify this, l.181 to 182: *"Despite the simulated increases in SAT and precipitation, δ18O simulated by HadCM3 shows a very weak trend of 0.04±0.003 ‰ per decade (r=0.21) over the last 50 years."*

The regional trends calculated by Stenni et al. (2017) over the last 100 years were based on ice core measurements, analysed as unweighted 5y-binned anomalies. This is clarified in the text, l. 192: *"Stenni et al. (2017) made a δ18O trend statistics based on ice core anomalies using unweighted composites over the period 1900-2000, based on 5-years bins."*
This method was differs from Casado et al., 2023, which we believe used resampled sub-annual records, and annual means over a 60 years or less time-window. With data then stacked without independent renormalisation using the variance. Note that we did not compute these results, rather directly took them from Table 2 of Stenni et al. (2017). S5 section of the supplementary material in Casado et al 2023 attributes the differences between Stenni and Casado papers to the different time windows (100 years for A2k against 35-45 years in the Casado paper).

Finally, we are very aware of the added value that brought by Casado et al 2023, and revise this sentence to include water stable isotope enabled GCM, l.201 to 205: *"These disparities could be explained by the different time windows, the different methodologies, the lack of ice core data to make representative regional reconstructions, or a model discrepancy. While Casado et al. (2023) carefully investigated the impact of the data stack method and the time-window on the δ18O reported trends, we suggest that an extended study could compare the statistical and dynamical methods on both ice core data and water stable isotope enabled GCM outputs to complete the analysis."*

Lines 169 to 170: "Before this, we provide a brief overview of the regional picture. At the regional scale, over the Historical period, trends are small (Figure 2)."
The regions in Figure 2 seem to be different than the ones in Stenni et al 2017 beyond the impact that the model grid would do to the attribution. For instance, none of the coastal grid points are included in your analysis, which differs strongly from Stenni et al. Victoria Land region extend further east near the coast (with the strong consequence of adding an additional core in this grid point compared to Stenni et al). Another notable difference comes from the lack of the coast part in DML coast region, which means that almost none of the ice core available for this region are represented in your average. As this seems to be a significant difference compared to Stenni et al, 2017, where several pages were included to explain the choice of the region, it needs to be justified.

To compute regional results, S.O used exactly the same code that she used for the Stenni et al. (2017) study. The notable differences come from the low resolution of the HadCM3 model (2.75° x 3.5°, lat x lon), which explains the highest differences on coastal areas. We agree this is indeed a significant limitation, as reported l.307.

Lines 171 to 172: "and is the highest for the Weddell coast with a trend of 0.05 ‰ per decade (r=0.39), and the strongest for the peninsula with a trend of 0.04 ‰ per decade (r=0.57)."
Which one is it ? The "highest" and the "strongest" should be the same.

High/low refers to gradients of the linear regressions while strong/weak refers to the correlation coefficient of the linear regressions. This is now clarified, l.187 to 190: *"In terms of linear relationship, it is null for the Victoria Land, while the gradient is the highest for the Weddell coast with a trend of 0.05 ‰ per decade (r=0.39), and the correlation coefficient is the highest (e.g. the strongest linear relationship) for the peninsula with a trend of 0.04 ‰ per decade (r=0.57)."*

Lines 183 to 185: "These disparities could be explained by the different time windows, the different methodologies or the lack of ice core data to make representative regional reconstructions." All of these hypotheses can be readily examined to ensure the robustness of the arguments detailed here. For the first hypothesis, it might be beneficial to incorporate a table in the supplementary materials, offering a comparison for the same time windows. Adaptations in methodologies can be explored, and Supplementary Table S3 in Casado et al., 2023, already presents trends using both their approach and the one from Stenni et al., 2017. Additionally, the absence of ice cores to establish representative regional reconstructions can be tested by focusing solely on specific grid points of the model corresponding to the ice core locations, comparing them to the regional average encompassing all grid points in the region. A fourth option, which the author does not explicitly address, is the potential bias or insufficient representation of variability in HadCM3, as suggested by Casado et al., 2023, particularly for most CMIP models.

This is another good point. The possibility that model may have insufficient representation of variability in HadCM3, as suggested by Casado et al., 2023, alongside the other possible reasons for the discrepancies outlined by the reviewer are briefly added at l.201: *"These disparities could be*

*explained by the different time windows, the different methodologies, the lack of ice core data to make representative regional reconstructions, or a model discrepancy."*

While recomputing our regional $\delta^{18}O$ trends, we have also taken the opportunity to check significance and include that several of these regional (HadCM3) trends are not significant (at p-value>0.05): over the last 50 years. Only three regions, the Indian, the Weddell and the Dronning Maud Land coastal regions display significant linear relationships. Figure 2 is adapted by shading in grey non-significant trends and amending the caption: *"Grey shaded rows correspond to non significant relationships (p-value>0.05)."* In the text, we removed l. 185 to 187: *"Over the last fifty years, a part from the Victoria Land where a very weak trend appear, other regions present weaker trends with correlation coefficients now ranging from 0.11 to 0.38 while gradients increase with values ranging from 0.03 ‰ per decade for the WAIS and the plateau, to 0.14 ‰ per decade for the Weddell coast."*

And instead l.190 to 192 now read: *"Over the last fifty years, only three regions, the Indian, the Weddell and Dronning Maud Land coastal regions keep on displaying significant δ18O trends, that double or more compared to the Historical period, with gradients of 0.08, 0.08 and 0.14 ‰ per decade respectively."* Also, we adapted the comparison with the results from Casado et al. (2023): *"They found gradients with the same range of values, from 0.09 ‰ for the Indian coast, to 0.19 ‰ for the Weddell coast, while they found significant relationships where we do not, for time windows varying from 40 to 65 years. Note that for most of the regions, the significance of our simulated relationships disappear for time windows shorter than 75 years (Appendix D). This could be explained either by the simulated anthropogenic variability being too low, as suggested by (Casado et al., 2023), or a change of the drivers on δ18O."*

As suggested, we looked at the impact of the window length on our simulated regional $\delta^{18}O$ trends. Results were added in Appendix D. Except for Dronning Maud Land and the Weddell coast, we do not obtain higher gradients, but we observe, as written above, that most of the relationships become insignificant when taking window length are shorter than 75 years. This can be explained either by un underestimation of the variability (although we find Antarctic-wide SAT trends consistent with observations), or a change in the main drivers of $\delta^{18}O$.

Finally, as reported in Stenni et al. (2017) and other studies, it is clear that $\delta^{18}O$ data remain sparse in some regions of Antarctica and that more data are needed for a more robust representativity.

Lines 194 to 206: This is an interesting discussion, but again, it fails to address the elephant in the room which is the model biases. The maps show areas with non-significant link between isotopic composition and temperature in the model, which seems sound and robust, but how does it compared to observations in the field? For instance, no correlation is found at the site of Vostok, where precipitation isotopic composition shows a significant correlation (R = 0.63, slope of 0.35 permil per degree) (Touzeau et al., 2016), while the map suggest a slope of 0 with non-significant correlation. The slope and correlation at Dome C also seems lower (below 0.3 with a r <0.3) than in observations (R2 = 0.63 and slope of 0.49) (Stenni et al., 2016). In general, any discussions which could support the validity of the model outputs would strengthen the manuscript, or at least provide confidence interval on the range of values that can actually be interpreted.

Fully agreed, these sentences are therefore modified to also reflect similar observational - as well as model - results, l.220 to 223: *"Non significant relationships were also reported in observations and model outputs. For instance, Goursaud et al. (2018) report no SAT-δ18O relationship at the annual scale over the coast of Dronning Maud Land, the Victoria Land, some of the Indian coast and the Peninsula. An absence of SAT-δ18O relationship derived from firn/ice cores were also published (e.g. Goursaud et al., 2019; Bertler et al., 2011; Vega et al., 2016; Goursaud et al., 2017)."*

Lines 211: "Here, and also for other warm climate results" Is it actually relevant? The manuscript is about historical reconstructions. This could potentially be discussed in the end of the discussion, but this subsection feels like results.

*Again fully agreed, that part of the sentence is removed, l.235: "Here, we suggest this is mainly driven by sea ice retreat (See section 3.3)."*

Lines 219 to 220: "Interestingly, however, this is not the case when comparing between the last 50 years of our HadCM3 simulation and the ECHAM5-wiso simulation." It is obsolete to compare your result with yet another CMIP5 model, when the isotope enabled version ECHAM6 wiso is available for more than 3 years.

*Addressed. Please see the response (above) related to the replacement ECHAM5-wiso results from Stenni et al. (2017) with ECHAM6-wiso results.*

Lines 225 to 227: "all the historical SAT-d18O relationships are different from the LGM-PI ECHAM5, and LIG-PI HadCM3 relationships: Werner et al. (2018) report LGM-PI regional gradients in ECHAM5 that are 17-26% lower, while (Sime et al., 2009b) and (Holloway et al., 2016) present LIG-PI regional gradients that are ~50% lower for HadCM3." Nobody would expect the historical and the LGM-PI relationships to be the same, considering the difference of time scales and underlying mechanisms driving the temperature changes. Is this really necessary in this manuscript which is about historical changes in the isotope-temperature relationship?

*Sentence removed.*

Lines 228: "ECHAM5 towards ERA-40 reanalysis," There are two generations of newer ERA products. It is unclear why the authors did not use either of these products. If the nudging is not conducted with the newer products in a revised version of the manuscript, which is what is really needed here, a clear justification for this omission will be required.

*The first version of this manuscript used the same model runs as in Stenni et al. (2017), just to have something to compare to HadCM3. These were used from 1979 – 2013 to exclude the observed SAT bias before 1979 (before the assimilation of satellite data in the reanalyses; Goursaud et al., 2018). From 1979 the model was nudged towards ERA-interim. We corrected l.235 : « towards ERA-interim. »*

Lines 238 to 239: "The primary mechanism driving continental-scale SAT-d18O decoupling is the simulated loss of sea ice during the historical period (Figure 5DH)." The rationale that could explain how to make this assessment is not supported by Figure 4D to H. The patterns of sea ice concentration anomalies does not explain any link with the variations of temperature and isotopic composition inside Antarctica by itself. There is no correlation provided, no mechanism, no simulations in which the sea ice concentration is artificially varied to support this assessment. Overall, this entire section falls short in establishing any form of causality and warrants a comprehensive revision. The conclusion should be revised once the rest of the manuscript has been reassessed.

*The section on drivers of d18O was indeed one of the more tricky parts of the manuscript to construct. We fully agree with the reviewer that Fig4 alone does not make the case that loss of sea ice is a key driver of d18O, and also that the previous draft was not particularly well written in places, with key references to Figure 7 missing. This section is now re-written as follows:*

*"We use two approaches to investigate the mechanisms driving simulated δ18O changes. First, we separate and compare extreme warm and cold years both for annual (Figure 4, Table C1) and seasonal (Figure 5) data by generating (annual and seasonal) composites with mean annual Antarctic SAT anomalies greater than plus or minus two standard deviations from the mean, respectively. Second, we isolate the impact of changing precipitation seasonality on δ18O, showing simple months values (Figure 6) and also following the decomposition method used in Liu and Battisti (2015); Holloway et al. (2016) and Sime et al. (2019) (Figure 7). As expected, the spatial patterns of SAT, and sea ice anomalies tend to vary together, with the pattern is approximately mirrored between cold and warm composites (Figure 4, top and bottom panels, respectively). Whilst fully isolating the drivers of δ18O is tricky, together Fig 4 to 7 suggest that the primary mechanism driving continental-scale SAT-δ18O decoupling in HadCM3 is the simulated loss of sea ice over the historical period (Figure 5dh).*

*The September average sea ice area across the warm composite is 5.8 x106 km2 less than in the cold composite. Given that this reduction occurs primarily during winter (Figure 5c; there is almost no summertime sea ice around Antarctica), warmer years tend to receive relatively more precipitation during winter months compared to cold years, partially offsetting the warming signal in δ18O. This can be seen in Figure 5, displaying seasonal anomalies (for the winter season, e.g. from June to August, and for the summer season, e.g. from December to February) in precipitation, δ18O and sea ice between the warm and cold composites: the largest (smallest) precipitation and δ18O anomalies occur during winter (summer) months. Precipitation anomalies peak in autumn and winter, whilst δ18O anomalies peak in winter and spring (Figure 6), the latter coincident with the annual maximum sea ice extent and largest sea ice area anomalies. The relative increase in winter precipitation during warm years acts to reduce δ18O across Antarctica, compared to if the seasonality of precipitation remained unchanged. This is perhaps clearest seen in Fig. 7, where Fig. 7a is predominantly blue - which says that precipitation seasonality changes are acting to decrease δ18O. The effect of changing seasonality is particularly large in the Indian, Dronning Maud Land and Victoria Land (through the Wilkes Land) sectors, which are prone to air mass intrusions (Fig. 5c and 7a)."*

---

## Author Response (AR2)

First of all, I apologize for the delay with the decision on your manuscript.
Based on the two reviews of your paper, I suppose that it can be published after some major revisions.
We thank you for the time you dedicated to the review of our paper. In the following of this letter, we detailed the change we brought to the paper, consistently with our responses to reviewers as well as your comments.

In particular, please extend the discussion of different isotope-temperature trends obtained in your work and in the paper by Casado et al. (2023).
We clarified the different isotope-temperature trends obtained in our work and in the paper by Casado et al. (2023), p.6 l.182 to l.185:
*« Casado et al. (2023) provide a higher trend from 1950–2005 of 0.11±0.02 ‰ per decade, based on ice core data. Different reasons could explain that mismatch that we are not able to elucidate so far, inter alia: (i) a model discrepancy to resolve processes, (ii) the model resolution, (iii) the geographical distribution of the ice core locations, (iv) the different methods for the SAT – δ18O calibration. »*

We also opened the discussion by giving results from published observations : l.220 to 223: *"Non significant relationships were also reported in observations and model outputs. For instance, Goursaud et al. (2018) report no SAT-δ18O relationship at the annual scale over the coast of Dronning Maud Land, the Victoria Land, some of the Indian coast and the Peninsula. An absence of SAT-δ18O relationship derived from firn/ice cores were also published (e.g. Goursaud et al., 2019; Bertler et al., 2011; Vega et al., 2016; Goursaud et al., 2017)."*

If one of the reasons for this discrepancy is "different methods for the SAT – d18O calibration" than is it possible to judge which method is preferable?
We completed the different possible reasons for the discrepancies in the obtained trends, l.201 :
*"These disparities could be explained by the different time windows, the different methodologies, the lack of ice core data to make representative regional reconstructions, or a model discrepancy."*.

While we cannot check the effect of the grid size as our simulations were all run with a same grid size, we checked the impact of the window length on our simulated regional $\delta^{18}O$ trends. The results were integrated in Appendix D.

Please also address the other issues raised by Reviewer 2:
We reported our responses to the second reviewer relative to the four below points you focus on.

1) add a new careful evaluation of your HadCM3 model;
We have added a new careful evaluation of the model in Appendix A, evaluating the simulated Antarctic Surface Air Temperature (SAT), precipitation (P) and precipitation weighted $\delta^{18}O$. The results show as expected a warm bias in the Antarctic interior – this is also observed in other models such as in Polar WRF (Zhang et al., 2022); and a dry bias in coastal regions. Overall HadCM3 performs roughly in line with expectations derived from other similar models, and have a reasonable representation of Antarctic surface climate and $\delta^{18}O$.

In the text, we referred to the appendix l.94 to l.96: *"HadCM3 provides a reasonable representation of Antarctic climate and δ18O (Appendix A, as well as Turner et al., 2006; Tindall et al., 2009; Holloway et al., 2016)."*
Appendix A can be found from page 26:

*Appendix A: HadCM3 evaluation of Antarctic surface climate and δ18O*

**A1 Method**

[revised manuscript text omitted]

2) provide comparison with ECHAM6 rather than with ECHAM5;

We replaced the analysis with the latest generation of the AGCM ECHAM equipped with water isotopes: ECHAM6-wiso (Stevens et al., 2013, Cauquoin et al., 2019). As stated by the reviewer, compared to ECHAM5-wiso, the performance of the water isotopes in ECHAM6-wiso is clearly improved. This is attributed to: (i) a modification of the supersaturation parameters ; (ii) that the kinetic fractionation at the evaporation over oceans is now assumed to be independent of the wind speed in order to better represent the d-excess versus deuterium relationship from the Antarctic Snow reported by Masson-Delmotte et al. (2008) ; and finally (iii) that the sublimation processes now accounts for the isotopic content of snow over sea ice. Based on the evaluation of global simulations against ERA-interim and ERA5 reanalyses, Cauquoin and Werner (2021) report that the nudging does not significantly change the simulated isotope values, while increasing the resolution generally improves the performance of the simulations. However, the evaluation of the simulated water stable isotopes in precipitation over Antarctica remains rather qualitative (Figure 1, Cauquoin and Werner, 2021).

Having obtained this new model output data from the newer version of ECHAM, we performed the same analysis as previously applied to ECHAM5 and HadCM3. As implied by the reviewer, using the newer version of the ECHAM indeed entirely resolve the discrepancy between the models – ECHAM6-wiso and HadCM3 (in the newer ECHAM6 version) now have equivalent SAT-$\delta^{18}$O surface air temperature relationships.

We thus made the following changes in the text:
- In section 2 ("Data and methods"), l.106 to 114:
*"Our Historical SAT–δ18O linear relationship at the regional scale are compared with the regional slopes and correlation coefficients that we computed from the AGCM ECHAM6-wiso equipped with water stable isotopes (Cauquoin et al., 2019). The water stable module of this last generation of the model ECHAM was updated compared to its predecessor, especially (i) the supersaturation parameters, (ii) the kinetic fractionation at the evaporation over oceans, now assumed to be independent of the wind speed in order to better represent the d-excess versus deuterium relationship from the Antarctic Snow reported by (Masson-Delmotte et al., 2008), and finally (iii) the sublimation processes now accounting for the isotopic content of snow over sea ice. Here, we use a simulation run at a T127L95 resolution ( 0.9° x 0.9° horizontal resolution and 95 vertical*

*levels) and nudged towards the ERA5 reanalyses (Hersbach et al., 2020) over the period 1979 – 2022 Cauquoin and Werner (2021)."*

- In section 4 ("Temperature versus δ18O relationships"), l.211:
*"To enable a consideration of model dependency, we also compare our Historical ensemble against a nudged ECHAM6-wiso simulation (Table 1)."*

- In section 4.2 ("Stability over the Historical period and model dependancy"), l.242 to l.253:
*"Interestingly, the ECHAM6-wiso simulation and the last 50 years of our HadCM3 simulation display similar SAT-δ18O relationships. ECHAM6-wiso simulates slightly stronger relationships with a mean correlation coefficient difference of 0.04, while gradients tend to be slightly higher in HadCM3 with a gradient difference of 0.13 ‰/°C. The only notable differences are for Dronning Maud Land and the Indian coast with stronger relationships and higher gradients simulated by HadCM3 (Table 1). Thus, whilst it is unclear whether the nudging of ECHAM6 towards ERA5 reanalysis, the model resolutions or differences in sea ice behaviours, are the main reason for these discrepancies, it is clear that simulated temperature versus δ18O relationships have low but significant uncertainties. These need to be considered, both regionally and for the most relevant climate state, before being undertaking any inferences of past temperatures using isotopes measured in ice cores."*

In section 6 ("conclusions"),  l.307 to 308:
*"Interestingly, we find similar but slightly weaker SAT-δ18O correlations and slightly higher gradients compared to ERA5 –nudged ECHAM6-wiso simulations at the regional scale."*

Table 1 is updated to reflect the replacement of ECHAM5 with ECHAM6 output.

**Table 1. Historical SAT–$\delta^{18}$O relationships at the regional scale.** Slope (in ‰/°C) plus or minus the standard error, and the correlation coefficient (into brackets) of the surface-weighed average of surface air temperature against the surface-weighed average of $\delta^{18}$O for the Antarctic regions as defined in the PAGES Antarctica2k project (Stenni et al., 2017): the plateau, the Indian coast, the Weddell coast, the Peninsula, the WAIS, Victoria Land and Dronning Maud Land, simulated by the ECHAM6-wiso model (, over the period 1979-2022, 44 points, 'ECHAM6-wiso') and simulated by HadCM3 over he last 50 years (1955-2004, 50 points, 'last 50 years of HadCM3'), and over the whole historical simulated period (1851-2004, 154 points, 'Historical HadCM3') using the ensemble mean of the six simulations (see methods). All the relationships are significant (p-values<0.05).

| | ECHAM6-wiso | last 50 years of HadCM3 | Historical HadCM3 |
|---|---|---|---|
| Plateau | 0.48±0.07 [0.71] | 0.61±0.14 [0.52] | 0.57±0.07 [0.53] |
| Indian coast | 0.29±0.08 [0.48] | 0.55±0.15 [0.46] | 0.67±0.07 [0.59] |
| Weddell coast | 0.49±0.11 [0.57] | 0.57±0.11 [0.59] | 0.57±0.07 [0.57] |
| Peninsula | 0.37±0.05 [0.74] | 0.28±0.06 [0.52] | 0.31±0.02 [0.71] |
| WAIS | 0.56±0.07 [0.75] | 0.60±0.12 [0.58] | 0.50±0.05 [0.61] |
| Victoria Land | 0.43±0.13 [0.46] | - | 0.30±0.12 [0.19] |
| Dronning Maud Land | 0.43±0.13 [0.46] | 0.76±0.12 [0.69] | 0.49±0.05 [0.60] |
| West Antarctica | 0.49±0.11 [0.59] | 0.50±0.10 [0.57] | 0.70±0.07 [0.62] |
| East Antarctica | 0.48±0.08 [0.69] | 0.49±0.10 [0.57] | 0.56±0.06 [0.58] |
| All Antarctica | 0.45±0.09 [0.59] | 0.67±0.13 [0.60] | 0.57±0.06 [0.62] |

3) extend the discussion of the SAM impact on the isotope signal;
new analysis of the impact of the SAM is given in Appendix G. This shows that HadCM3 reproduces the impacts of SAM on SAT and P reported in previous studies (Clem et al., 2016; Fogt et al., 2020), *i.e.* colder and drier conditions in a positive SAM. For $\delta^{18}$O, HadCM3 simulates depletion in most areas of the Antarctic continent while the SAM is in a positive phase, but these

results are associated with relatively low correlation coefficients with means of -0.26±0.11 over the Historical period and -0.27±0.12 for the period 1950 – 2004. We thus conclude that our simulations cannot establish a robust link between the SAM and the Antarctic precipitation weighted $\delta^{18}$O. This result is supported by the diversity of $\delta^{18}$O measurements from precipitation and firn/ice cores on different Antarctic locations (*e.g.* Vega et al., 2016; Kino et al., 2021; Servettaz, 2022; Dreossi et al., 2023). Moreover, it was shown that SAM impacts are different with the ENSO phases (Wilson et al., 2016), and that other modes affect Antarctic climate (*e.g.* Shields et al., 2022). Further analysis on the impact of the atmospheric circulation on Antarctic precipitation weighted $\delta^{18}$O for the Historical period would need to be the subject of a future study. The new results are references in Section 5 ("Drivers") p10 l.292 to l.295 as:

*"The dynamic processes behind the sea ice extent induced δ18O changes are complex and multiple. Although the Southern Annular Mode, leading mode of the atmospheric variability in the Southern Hemisphere, might explain part of these δ18O simulated changes (Appendix G), a more comprehensive study might investigate the impact of the atmospheric circulation changes."*

In the conclusion, l.310, we replaced:
*"We identify three processes [...]"* by *"We suggest [...]"*, meaning that an extended study is necessary to check the atmospheric processes at the origin of our simulated results.

Here is our new Appendix G:

[revised manuscript text omitted]

4) better describe the model and the simulations setup, and, in particular, was the sea ice simulated or prescribed? (also requested by the first Reviewer);

The paragraph is re-ordered as requested. Parts relating to CMIP6 are removed, to prevent confusion, and instead the details of the protocol are given, as requested, after the model description. This includes, as requested some more information on the HadCM3 coupled Atmosphere-Ocean model, and that sea-ice is not prescribed but calculated. The paragraph dedicated to our model description and simulations is now p.3 l.79 to 90:

*"Here, we use the Hadley Center Atmosphere-Ocean general circulation model (HadCM3; AOGCM), to run six transient Historical simulations. HadCM3 is a version of the coupled Atmosphere-Ocean UK Met Office climate model (Pope et al., 2000; Gordon et al., 2000), which means that sea ice is prognostic. The model is equipped with stable water isotopes (Tindall et al., 2009). Its horizontal resolution is 3.75° × 2.5°, and there are 19 vertical levels (Pope et al., 2000; Gordon et al., 2000; Tindall et al., 2009). The setup of the Historical simulations is described in (Schurer et al., 2014), and follows the recommendations of the third Paleoclimate Modelling Intercomparison Project (PMIP3; Schmidt et al., 2011)(PMIP3; Schmidt et al. 2012). Each simulation is forced with time-varying orbital, solar, volcanic, land-use and well-mixed greenhouse gas forcing. As above, sea ice is not prescribed, rather calculated by the model. Changes in orbital parameters were calculated following (Berger, 1978). Volcanic forcing is that described in (Crowley et al., 2008). The solar forcing follows (Shapiro et al., 2011). Changes in CO2, N2O and CH4 were set following the PMIP3 standard (Schmidt et al., 2011). Changes in the abundances of 6 Halocarbons were prescribed following (Tett et al., 2007). Changes in land-cover were prescribed by reclassifying the Global land cover reconstruction developed by (Pongratz et al., 2008). Each of our simulations were only altered by starting each simulation a year apart."*

5) address a number of minor comments and questions.

We integrated the changes we reported in our responses to the reviewers'minor comments, as spotted in the track-change version.

Also, as requested by Reviewer 1, please improve the figures, in particular Figure 2.

Finally, we improved the figures increasing label and title sizes. In Figure 2, we tried to simplify the reading by relocating the subplots and adding the names of the regions.